# Revisiting Realistic Test-Time Training: Sequential Inference and Adaptation by Anchored Clustering

**Yongyi Su**[1*]   **Xun Xu**[21*†]   **Kui Jia**[13†]

[1]South China University of Technology    [2]Institute for Infocomm Research
[3]Peng Cheng Laboratory
`eesuyongyi@mail.scut.edu.cn`
`alex.xun.xu@gmail.com`
`kuijia@scut.edu.cn`

## Abstract

Deploying models on target domain data subject to distribution shift requires adaptation. Test-time training (TTT) emerges as a solution to this adaptation under a realistic scenario where access to full source domain data is not available and instant inference on target domain is required. Despite many efforts into TTT, there is a confusion over the experimental settings, thus leading to unfair comparisons. In this work, we first revisit TTT assumptions and categorize TTT protocols by two key factors. Among the multiple protocols, we adopt a realistic sequential test-time training (sTTT) protocol, under which we further develop a *test-time anchored clustering (TTAC)* approach to enable stronger test-time feature learning. TTAC discovers clusters in both source and target domain and match the target clusters to the source ones to improve generalization. Pseudo label filtering and iterative updating are developed to improve the effectiveness and efficiency of anchored clustering. We demonstrate that under all TTT protocols TTAC consistently outperforms the state-of-the-art methods on six TTT datasets. We hope this work will provide a fair benchmarking of TTT methods and future research should be compared within respective protocols. A demo code is available at `https://github.com/Gorilla-Lab-SCUT/TTAC`.

## 1   Introduction

The recent success in deep learning is attributed to the availability of large labeled data [15, 40] and the assumption of i.i.d. between training and test datasets. Such assumptions could be violated when test data features a drastic difference from the training data, e.g. training on synthetic images and test on real ones, and this is often referred to as domain shift [22, 2]. To tackle this issue, domain adaptation (DA) [32] emerges and the labeled training data and unlabeled testing data are often referred to as source and target data/domains respectively.

The existing DA works either require the access to both source and target domain data during training [5] or training on multiple domains simultaneously [39]. The former approach renders the methods restrictive to limited scenarios where source domain data is always available during adaptation while the latter ones are computationally more expensive. To alleviate the reliance on source domain data, which may be inaccessible due to privacy issues or storage overhead, source-free domain adaptation (SFDA) emerges which handles DA on target data without access to source data [19, 16, 37, 35, 20]. SFDA is often achieved through self-training [19], self-supervised

---

*Equal contribution

†Correspondence to <kuijia@scut.edu.cn> & <alex.xun.xu@gmail.com>

learning [20] or introducing prior knowledge [19] and it requires multiple training epochs on the full target data to allow convergence. Despite easing the dependence on source data, SFDA has major drawbacks in a more realistic domain adaptation scenario where test data arrives in a stream and inference or prediction must be taken instantly, and this setting is often referred to as test-time training (TTT) or adaptation (TTA) [27, 31, 13, 20]. Despite the attractive feature of adaption at time test, we notice a confusion of what defines a test-time training and as a result comparing apples and oranges happens frequently in the community. In this work, we first categorize TTT by two key factors after summarizing various definitions made in existing works. First, under a realistic TTT setting, test samples are sequentially streamed and prediction must be made instantly upon the arrival of a new test sample. More specifically, the prediction of test sample $X_T$, arriving at time stamp $T$, should not be affected by any subsequent samples, $\{X_t\}_{t=T+1\cdots\infty}$. Throughout this work, we refer to the sequential streaming as **one-pass adaptation** protocol and any other protocols violating this assumption are called **multi-pass adaptation** (model may be updated on all test data for multiple epochs before inference). Second, we notice some recent works must **modify source domain training loss**, e.g. by introducing additional self-supervised branch, to allow more effective TTT [27, 20]. This will introduce additional overhead in the deployment of TTT because re-training on some source dataset, e.g. ImageNet, is computationally expensive. In this work, we aim to tackle on the most realistic and challenging TTT protocol, i.e. one-pass test time training with no modifications to training objective. This setting is similar to TTA proposed in [31] except for not restricting access to a light-weight information from the source domain. Given the objective of TTT being efficient adaptation at test-time, this assumption is computationally efficient and improves TTT performance substantially. We name this new TTT protocol as **sequential test time training (sTTT)**.

We propose two techniques to enable efficient and accurate sTTT. i) We are inspired by the recent progresses in unsupervised domain adaptation [28] that encourages testing samples to form clusters in the feature space. However, separately learning to cluster in the target domain without regularization from source domain does not guarantee improved adaptation [28]. To overcome this challenge, we identify clusters in both the source and target domains through a mixture of Gaussians with each component Gaussian corresponding to one category. Provided with the category-wise statistics from source domain as anchors, we match the target domain clusters to the anchors by minimizing the KL-Divergence as the training objective for sTTT. Therefore, we name the proposed method *test-time anchored clustering (TTAC)*. Since test samples are sequentially streamed, we develop an exponential moving averaging strategy to update the target domain cluster statistics to allow gradient-based optimization. ii) Each component Gaussian in the target domain is updated by the test sample features that are assigned to the corresponding category. Thus, incorrect assignments (pseudo labels) will harm the estimation of component Gaussian. To tackle this issue, we are inspired by the correlation between network's stability and confidence and pseudo label accuracy [17, 24], and propose to filter out potentially incorrect pseudo labels. Component Gaussians are then updated by the samples that have passed the filtering. To exploit the filtered out samples, we incorporate a global feature alignment [20] objective. We also demonstrate TTAC is compatible with existing TTT techniques, e.g. contrastive learning branch [20], if source training loss is allowed to be modified. The contributions of this work are summarized as below.

- In light of the confusions within TTT works, we provide a categorization of TTT protocols by two key factors. Comparison of TTT methods is now fair within each category.

- We adopt a realistic TTT setting, namely sTTT. To improve test-time feature learning, we propose TTAC by matching the statistics of the target clusters to the source ones. The target statistics are updated through moving averaging with filtered pseudo labels.

- The proposed method is complementary to existing TTT method and is demonstrated on six TTT datasets, achieving the state-of-the-art performance under all categories of TTT protocols.

## 2 Related Work

**Unsupervised Domain Adaptation**. Domain adaptation aims to improve model generalization when source and target data are not drawn i.i.d. When target data are unlabeled, unsupervised domain adaptation (UDA) [5, 29] learns domain invariant feature representations on both source and target domains to improve generalization. Follow-up works improve UDA by minimizing a divergence [6, 25, 38], adversarial training [11] or discovering cluster structures in the target data [28].

Apart from formulating UDA as a task-specific model, re-weighting has been adopted for domain adaptation by selectively up-weighting conducive samples in the source domain [14, 36]. During model training, the existing approaches often require access to the source domain data which, however, may be not accessible due to privacy issues, storage overhead, etc. Therefore, deploying UDA in more realistic scenarios has inspired research into source-free domain adaptation and test-time training/adaptation.

**Source-Free Domain Adaptation**. Without the access to source data, source-free domain adaptation (SFDA) develops domain adaptation through self-training [19, 16, 13], self-supervised training [20], clustering in the target domain [37] and feature restoration [4]. It has been demonstrated that SFDA performs well on seminal domain adaptation datasets even compared against UDA methods [28]. Nevertheless, SFDA requires access to all testing data beforehand and model training must be carried out iteratively on the testing data. In a more realistic DA scenario where inference and adaptation must be implemented simultaneously, SFDA will no longer be effective. Moreover, some statistical information on the source domain does not pose privacy issues and can be exploited to further improve adaptation on target data.

**Test-Time Training**. Collecting enough samples from target domain and adapt models in an offline manner restricts the application to adapting to a static target domain. To allow fast and online adaptation, test-time training (TTT) [27, 33] or adaptation (TTA) [31] emerges. Despite many recent works claiming to be test-time training, we notice a severe confusion over the definition of TTT. In particular, whether training objective must be modified [27, 20] and whether sequential inference on target domain data is possible [31, 13]. Therefore, to reflect the key challenges in TTT, we define a setting called sequential test-time training (sTTT) which neither modifies the training objective nor violates sequential inference. Under the more clear definition, some existing works, e.g. TTT [27] and TTT++ [20] is more likely to be categorized into SFDA. Several existing works [31, 13] can be adapted to the sTTT protocol. Tent [31] proposed to adjust affine parameters in the batchnorm layers to adapt to target domain data. Nevertheless, updating only a fraction of model weights inevitably leads to limited performance gain on the target domain. T3A [13] further proposed to update classifier prototype through pseudo labeling. Despite being efficient, updating classifier prototype alone does not affect feature representation for the target domain. Target feature may not form clusters at all when the distribution mismatch between source and target is large enough. In this work we propose to simultaneously cluster on the target domain and match target clusters to source domain classes, namely anchored clustering. To further constrain feature update, we introduce additional global feature alignment and pseudo label filtering. Through the introduced anchored clustering, we achieve test-time training of more network parameters and achieve the state-of-the-art performance.

## 3 Methodology

In this section we first introduce the anchored clustering objective for test-time training through pseudo labeling and then describe an efficient iterative updating strategy. An overview of the proposed pipeline is illustrated in Fig. 1.

### 3.1 Anchored Clustering for Test-Time Training

Discovering cluster structures in the target domain has been demonstrated effective for unsupervised domain adaptation [28] and we develop an anchored clustering on the test data alone. We first use a mixture of Gaussians to model the clusters in the target domain, here each component Gaussian represents one discovered cluster. We further use the distributions of each category in the source domain as anchors for the target distribution to match against. In this way, test data features can simultaneously form clusters and the clusters are associated with source domain categories, resulting in improved generalization to target domain. Formally, we first write the mixture of Gaussians in the source and target domains $p_s(x) = \sum_k \alpha_k \mathcal{N}(\mu_{sk}, \Sigma_{sk}), \quad p_t(x) = \sum_k \beta_k \mathcal{N}(\mu_{tk}, \Sigma_{tk})$, where $\{\mu_k \in \mathbb{R}^d, \Sigma_k \in \mathbb{R}^{d \times d}\}$ represent one cluster in the source/target domain and $d$ is the dimension of feature embedding.

Anchored clustering can be achieved by matching the above two distributions and one may directly minimize the KL-Divergence between the two distribution. Nevertheless, this is non-trivial because the KL-Divergence between two mixture of Gaussians has no closed-form solution which prohibits efficient gradient-based optimization. Despite some approximations exist [10], without knowing the

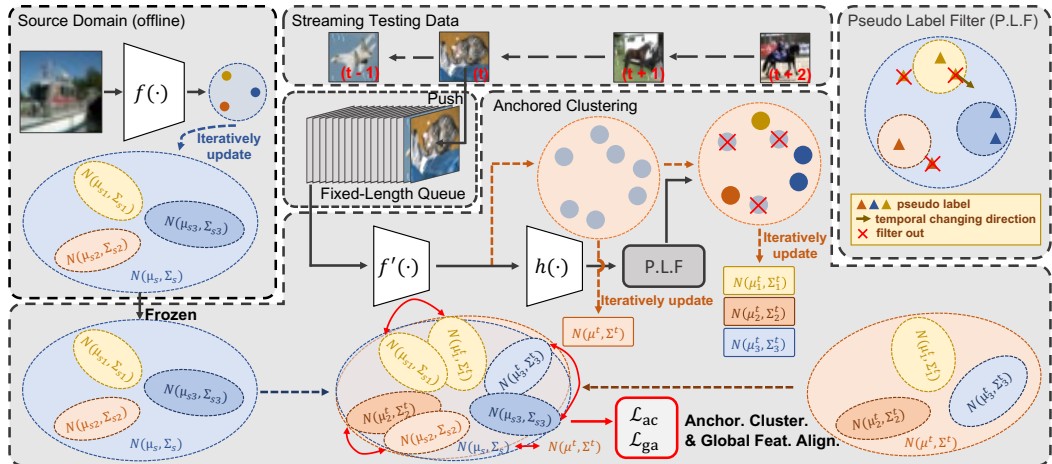

Figure 1: Overview of TTAC pipeline. i) In the source domain, we calculate category-wise and global statistics as anchors. ii) In the testing stage, samples are sequentially streamed and pushed into a fixed-length queue. Clusters in target domain are identified through anchored clustering with pseudo label filtering. Target clusters are then matched to the anchors in source domain to achieve test-time training.

semantic labels for each Gaussian component, even a good match between two mixture of Gaussians does not guarantee target clusters are aligned to the correct source ones and this will severely harm the performance of test-time training. In light of these challenges, we propose a category-wise alignment. Specifically, we allocate the same number of clusters in both source and target domains and each target cluster is assigned to one source cluster. We can then minimize the KL-Divergence between each pair of clusters as in Eq. 1.

$$
\begin{aligned}
\mathcal{L}_{ac} &= \sum_k D_{KL}(\mathcal{N}(\mu_{sk}, \Sigma_{sk}) || \mathcal{N}(\mu_{tk}, \Sigma_{tk})) \\
&= \sum_k -H(\mathcal{N}(\mu_{sk}, \Sigma_{sk})) + H(\mathcal{N}(\mu_{sk}, \Sigma_{sk}), \mathcal{N}(\mu_{tk}, \Sigma_{tk}))
\end{aligned}
\tag{1}
$$

The KL-Divergence can be further decomposed into the entropy $H(\mathcal{N}(\mu_{sk}, \Sigma_{sk}))$ and cross-entropy $H(\mathcal{N}(\mu_{sk}, \Sigma_{sk}), \mathcal{N}(\mu_{tk}, \Sigma_{tk}))$. It is commonly true that the source reference distribution $P_s(x)$ is fixed thus the entropy term is a constant $C$ and only the cross-entropy term is to be optimized. Given the closed-form solution to the KL-Divergence between two Gaussian distributions, we now write the anchored clustering objective as,

$$
\mathcal{L}_{ac} = \sum_k \{ \log \sqrt{2\pi^d |\Sigma_{tk}|} + \frac{1}{2}(\mu_{tk} - \mu_{sk})^\top \Sigma_{tk}^{-1}(\mu_{tk} - \mu_{sk}) + tr(\Sigma_{tk}^{-1}\Sigma_{sk}) \} + C
\tag{2}
$$

The source cluster parameters can be estimated in an offline manner. These information will not cause any privacy leakage and only introduces a small computation and storage overheads. In the next section, we elaborate clustering in the target domain.

### 3.2 Clustering through Pseudo Labeling

In order to test-time train network with anchored clustering loss, one must obtain target cluster parameters $\{\mu_{tk}, \Sigma_{tk}\}$. For a minibatch of target test samples $\mathcal{B}^t = \{x_i\}_{i=1...N_B}$ at timestamp $t$, we first denote the predicted posterior as $P^t = softmax(h(f(x_i))) \in [0, 1]^{B \times K}$ where $softmax(\cdot)$, $h(\cdot)$ and $f(\cdot)$ respectively denote a standard softmax function, the classifier head and backbone network. The pseudo labels are obtained via $\hat{y}_i = \arg\max_k P_{ik}^t$. Given the predicted pseudo labels we could estimate the mean and covariance for each component Gaussian with the pseudo labeled testing samples. However, pseudo labels are always subject to model's discrimination ability. The error rate for pseudo labels is often high when the domain shift between source and target is large, directly updating the component Gaussian is subject to erroneous pseudo labels, a.k.a. confirmation bias [1]. To reduce the impact of incorrect pseudo labels, we first adopt a light-weight temporal consistency (TC) pseudo label filtering approach. Compared to co-teaching [7] or meta-learning [18] based methods, this light-weight method does not introduce additional computation overhead and

is therefore more suitable for test-time training. Specifically, to alleviate the impact from the noisy predictions, we calculate the temporal exponential moving averaging posteriors $\widetilde{P}^t \in [0,1]^{N \times K}$ as below,

$$\widetilde{P}_i^t = (1 - \xi) * \widetilde{P}_i^{t-1} + \xi * P_i^t \quad , \quad s.t. \quad \widetilde{P}_i^0 = P_i^0 \tag{3}$$

The temporal consistency filtering is realized as in Eq. 4 where $\tau_{TC}$ is a threshold determining the maximally allowed difference in the most probable prediction over time. If the posterior deviate from historical value too much, it will be excluded from target domain clustering.

$$F_i^{TC} = \mathbb{1}((P_{i\hat{k}}^t - \widetilde{P}_{i\hat{k}}^{t-1}) > \tau_{TC}), \quad s.t. \quad \hat{k} = \arg\max_k(P_{ik}^t) \tag{4}$$

Due to the sequential inference, test samples without enough historical predictions may still pass the TC filtering. So, we further introduce an additional pseudo label filter directly based on the posterior probability as,

$$F_i^{PP} = \mathbb{1}(\widetilde{P}_{i\hat{k}}^t > \tau_{PP}) \tag{5}$$

By filtering out potential incorrect pseudo labels, we update the component Gaussian only with the leftover target samples as below.

$$\mu_{tk} = \frac{\sum_i F_i^{TC} F_i^{PP} \mathbb{1}(\hat{y}_i = k) f(x_i)}{\sum_i F_i^{TC} F_i^{PP} \mathbb{1}(\hat{y}_i = k)}, \quad \Sigma_{tk} = \frac{\sum_i F_i^{TC} F_i^{PP} \mathbb{1}(\hat{y}_i = k)(f(x_i) - \mu_{tk})^\top (f(x_i) - \mu_{tk})}{\sum_i F_i^{TC} F_i^{PP} \mathbb{1}(\hat{y}_i = k)} \tag{6}$$

### 3.3 Global Feature Alignment

As discussed above, test samples that do not pass the filtering will not contribute to the estimation of target clusters. Hence, anchored clustering may not reach its full potential without the filtered test samples. To exploit all available test samples, we propose to align global target data distribution to the source one. We define the global feature distribution of the source data as $\hat{p}_s(x) = \mathcal{N}(\mu_s, \Sigma_s)$ and the target data as $\hat{p}_t(x) = \mathcal{N}(\mu_t, \Sigma_t)$. To align two distributions, we again minimize the KL-Divergence as,

$$\mathcal{L}_{ga} = D_{KL}(\hat{p}_s(x)||\hat{p}_t(x)) \tag{7}$$

Similar idea has appeared in [20] which directly matches the moments between source and target domains [38] by minimizing the F-norm for the mean and covariance, i.e. $||\mu_t - \mu_s||_2^2 + ||\Sigma_t - \Sigma_s||_F^2$. However, designed for matching complex distributions represented as drawn samples, central moment discrepancy [38] requires summing infinite central moment discrepancies and the ratios between different order moments are hard to estimate. For matching two parameterized Gaussian distributions KL-Divergence is more convenient with good explanation from a probabilistic point of view. Finally, we add a small constant to the diagonal of $\Sigma$ for both source and target domains to increase the condition number for better numerical stability.

### 3.4 Efficient Iterative Updating

Despite the distribution for source data can be trivially estimated from all available training data in a totally offline manner, estimating the distribution for target domain data is not equally trivial, in particular under the sTTT protocol. In a related research [20], a dynamic queue of test data features are preserved to dynamically estimate the statistics, which will introduce additional memory footprint [20]. To alleviate the memory cost we propose to iteratively update the running statistics for Gaussian distribution. Formally, we define $t$-th test minibatch as $\mathcal{B}^t = \{x_i\}_{i=1 \cdots N_B}$. Denoting the running mean and covariance at step $t$ as $\mu^t$ and $\Sigma^t$, we present the rules to update the mean and covariance in Eq. 8. More detailed derivations and update rules for per cluster statistics are deferred to the Appendix.

$$\mu^t = \mu^{t-1} + \delta^t, \quad \Sigma^t = \Sigma^{t-1} + a^t \sum_{x_i \in \mathcal{B}} \{(f(x_i) - \mu^{t-1})^\top (f(x_i) - \mu^{t-1}) - \Sigma^{t-1}\} - \delta^{t\top} \delta^t$$

$$\delta^t = a^t \sum_{x_i \in \mathcal{B}} (f(x_i) - \mu^{t-1}), \quad N^t = N^{t-1} + |\mathcal{B}^t|, \quad a^t = \frac{1}{N^t} \tag{8}$$

Additionally, $N^t$ grows larger overtime. New test samples will have smaller contribution to the update of target domain statistics when $N^t$ is large enough. As a result, the gradient calculated from current minibatch will vanish. To alleviate this issue, we impose a clip on the value of $\alpha^t$ as below. As such, the gradient can maintain a minimal scale even if $N^t$ is very large.

$$a^t = \begin{cases} \frac{1}{N^t} & N^t < N_{clip} \\ \frac{1}{N_{clip}} & others \end{cases} \qquad (9)$$

### 3.5 TTAC Training Algorithm

We summarize the training algorithm for the TTAC in Algo. 1. For effective clustering in target domain, we allocate a fixed length memory space, denoted as $\mathcal{C} \in \mathbb{R}^{N_C \times H \times W \times 3}$, to store the recent testing samples. In the sTTT protocol, we first make instant prediction on each testing sample, and only update the model when $N_B$ testing samples are accumulated. TTAC can be efficiently implemented, e.g. with two devices, one is for continuous inference and another is for model updating.

---

**Algorithm 1:** Test-Time Anchored Clustering Training Algorithm

---

**input** : A new testing sample batch $\mathcal{B}^t = \{x_i\}_{i=1...N_B}$.
  # Update the testing sample queue $\mathcal{C}$.
$\mathcal{C}^t = \mathcal{C}^t \setminus \mathcal{B}^{t-N_C/N_B}, \quad \mathcal{C}^t = \mathcal{C}^t \bigcup \mathcal{B}^t$
**for** 1 to $N_{itr}$ **do**
    **for** *minibatch* $\{x_k^t\}_{k=1}^N$ *in* $\mathcal{C}^t$ **do**
        # Obtain the predicted posterior and pseudo labels
        $P_i^t = softmax(h(f(x_i^t))), \quad \hat{y}_i^t = \arg\max_k(P_{ik}^t)$
        # Calculate the global and per-cluster running mean and covariance by Eq. 8
        $\mu^t, \quad \Sigma^t, \quad \{\mu_k^t\}, \quad \{\Sigma_k^t\}$
        # Optimize the combined loss by Eq. 2 and Eq. 7
        $\mathcal{L} = \mathcal{L}_{ac} + \lambda\mathcal{L}_{ga}$
        update network $f$ to minimize $\mathcal{L}$

---

## 4 Experiment

In this section, we first compare various existing methods based on the two key factors. Evaluation is then carried out on six test-time training datasets. We then ablate the components of TTAC. Further analysis on the cumulative performance, qualitative insights, etc. are provided at the end.

### 4.1 Datasets

We evaluate on 6 test-time training datasets and report the classification error rate (%) throughout the experiment section. To evaluate the test-time training efficacy on corrupted target images, we use **CIFAR10-C/CIFAR100-C** [9], each consisting of 10/100 classes with 50,000 training samples of clean data and 10,000 corrupted test samples. We further evaluate test-time training on hard target domain samples with **CIFAR10.1** [23], which contains around 2,000 difficult testing images sampled over years of research on the original CIFAR-10 dataset. To demonstrate the ability to do test-time training for synthetic data to real data transfer we further use **VisDA-C** [21], which is a challenging large-scale synthetic-to-real object classification dataset, consisting of 12 classes, 152,397 synthetic training images and 55,388 real testing images. To evaluate large-scale test-time training, we use **ImageNet-C** [9] which consists of 1,000 classes and 15 types of corruptions on the 50,000 testing samples. Finally, to evaluate test-time training on 3D point cloud data, we choose **ModelNet40-C** [26], which consists of 15 common and realistic corruptions of point cloud data, with 9,843 training samples and 2,468 test samples.

### 4.2 Experiment Settings

**Hyperparameters**. We use the ResNet-50 [8] for image datasets and the DGCNN [34] on ModelNet40-C. We optimize the backbone network $f(\cdot)$ by SGD with momentum on all datasets. On CIFAR10-C/CIFAR100-C and CIFAR10.1, we use (batchsize) BS = 256 and (learning rate) LR = 0.01, 0.0001, 0.01 respectively. On VisDA-C we use BS = 128 and LR = 0.0001, and on ModelNet40-C we use BS = 64 and LR = 0.001. More details of hyperparameters can be found in the Appendix.

**Test-Time Training Protocols**. We categorize test-time training based on two key factors. First, whether the training objective must be changed during training on the source domain, we use Y and N

to indicate if training objective is allowed to be changed or not respectively. Second, whether testing data is sequentially streamed and predicted, we use O to indicate a sequential **O**ne-pass inference and M to indicate non-sequential inference, a.k.a. **M**ulti-pass inference. With the above criteria, we summarize 4 test-time training protocols, namely N-O, Y-O, N-M and Y-M, and the strength of the assumption increases from the first to the last protocols. Our sTTT setting makes the weakest assumption, i.e. N-O. Existing methods are categorized by the four TTT protocols, we notice that some methods can operate under multiple protocols

**Competing Methods**. We compare the following test-time training methods. Direct testing (**TEST**) without adaptation simply do inference on target domain with source domain model. Test-time training (**TTT-R**) [27] jointly trains the rotation-based self-supervised task and the classification task in the source domain, and then only train the rotation-based self-supervised task in the streaming test samples and make the predictions instantly. The default method is classified into the Y-M protocol. Test-time normalization (**BN**) [12] moving average updates the batch normalization statistics by streamed data. The default method follows N-M protocol and can be adapted to N-O protocol. Test-time entropy minimization (**TENT**) [31] updates the parameters of all batch normalization by minimizing the entropy of the model predictions in the streaming data. By default, TENT follows the N-O protocol and can be adapted to N-M protocol. Test-time classifier adjustment (**T3A**) [13] computes target prototype representation for each category using streamed data and make predictions with updated prototypes. T3A follows the N-O protocol by default. Source Hypothesis Transfer (**SHOT**) [19] freezes the linear classification head and trains the target-specific feature extraction module by exploiting balanced category assumption and self-supervised pseudo-labeling in the target domain. SHOT by default follows the N-M protocol and we adapt it to N-O protocol. **TTT++** [20] aligns source domain feature distribution, whose statistics are calculated offline, and target domain feature distribution by minimizing the F-norm between the mean covariance. TTT++ follows the Y-M protocol and we adapt it to N-O (removing contrastive learning branch) and Y-O protocols. Finally, we present our own approach, **TTAC**, which only requires a single pass on the target domain and does not have to modify the source training objective. We further modify TTAC for Y-O, N-M and Y-M protocols, for Y-O and Y-M we incorporate an additional contrastive learning branch [20]. We could further combine TTAC with additional diversity loss and entropy minimization loss introduced in SHOT [19], denoted as TTAC+SHOT.

## 4.3 Test-Time Training on Corrupted Target Domain

We present the test-time training results on CIFAR10/100-C and ModelNet40-C datasets in Tab. 1, and the results on ImageNet-C dataset in Tab. 2. We make the following observations from the results.

**sTTT (N-O) Protocol**. We first analyze the results under the proposed sTTT (N-O) protocol. Our method outperforms all competing ones by a large margin. For example, 3% improvement is observed on both CIFAR10-C and CIFAR100-C from the previous best (TTT++) and 5-13% improvement is observed on ImageNet-C compared with BN and TENT, and TTAC is superior in average accuracy and outperforms on 9 out of 15 types of corruptions compared with SHOT on ImageNet-C. We further combine TTAC with the class balance assumption made in SHOT (TTAC+SHOT). With the stronger assumptions out method can further improve upon TTAC alone, in particular on ModelNet40-C dataset. This result demonstrates TTAC's compatibility with existing methods.

**Alternative Protocols**. We further compare different methods under N-M, Y-O and Y-M protocols. Under the Y-O protocol, TTT++ [20] modifies the source domain training objective by incorporating a contrastive learning branch [3]. To compare with TTT++, we also include the contrastive branch and observe a clear improvement on both CIFAR10-C and CIFAR100-C datasets. More TTT methods can be adapted to the N-M protocol which allows training on the whole target domain data multiple epochs. Specifically, we compared with BN, TENT and SHOT. With TTAC alone we observe substantial improvement on all three datasets and TTAC can be further combined with SHOT demonstrating additional improvement. Finally, under the Y-M protocol, we demonstrate very strong performance compared to TTT-R and TTT++. It is also worth noting that TTAC under the N-O protocol can already yield results close to TTT++ under the Y-M protocol, suggesting the strong test-time training ability of TTAC even under the most challenging TTT protocol.

## 4.4 Additional Datasets

**TTT on Hard Samples**. CIFAR10.1 contains roughly 2,000 new test images that were re-sampled after the research on original CIFAR-10 dataset, which consists of some hard samples and reflects the

Table 1: Comparison under different TTT protocols. Y/N indicates modifying source domain training objective or not. O/M indicate one pass or multiple passes test-time training. C10-C, C100-C and MN40-C refer to CIFAR10-C, CIFAR100-C and ModelNet40-C datasets respectively. All numbers indicate error rate in percentage.

| Method | TTT Protocol | Assum. Strength | C10-C | C100-C | MN40-C |
|---|---|---|---|---|---|
| TEST | - | - | 29.15 | 60.34 | 34.62 |
| BN [12] | N-O | Weak | 15.49 | 43.38 | 26.53 |
| TENT [31] | N-O | Weak | 14.27 | 40.72 | 26.38 |
| T3A [13] | N-O | Weak | 15.44 | 42.72 | 24.57 |
| SHOT [19] | N-O | Weak | 13.95 | 39.10 | 19.71 |
| TTT++ [20] | N-O | Weak | 13.69 | 40.32 | - |
| TTAC (Ours) | N-O | Weak | **10.94** | 36.64 | 22.30 |
| TTAC+SHOT (Ours) | N-O | Weak | 10.99 | **36.39** | **19.21** |
| TTT++ [20] | Y-O | Medium | 13.00 | 35.23 | - |
| TTAC (Ours) | Y-O | Medium | **10.69** | **34.82** | - |
| BN [12] | N-M | Medium | 15.70 | 43.30 | 26.49 |
| TENT [31] | N-M | Medium | 12.60 | 36.30 | 21.23 |
| SHOT [19] | N-M | Medium | 14.70 | 38.10 | 15.99 |
| TTAC (Ours) | N-M | Medium | **9.42** | 33.55 | 16.77 |
| TTAC+SHOT (Ours) | N-M | Medium | 9.54 | **32.89** | **15.04** |
| TTT-R [27] | Y-M | Strong | 14.30 | 40.40 | - |
| TTT++ [20] | Y-M | Strong | 9.80 | 34.10 | - |
| TTAC (Ours) | Y-M | Strong | **8.52** | **30.57** | - |

Table 2: Test-time training on ImageNet-C under the sTTT (N-O) protocol.

| Method | Birt | Contr | Defoc | Elast | Fog | Frost | Gauss | Glass | Impul | Jpeg | Motn | Pixel | Shot | Snow | Zoom | Avg |
|---|---|---|---|---|---|---|---|---|---|---|---|---|---|---|---|---|
| TEST | 38.82 | 89.55 | 82.23 | 87.13 | 64.84 | 76.83 | 97.34 | 90.50 | 97.76 | 68.31 | 83.60 | 80.37 | 96.74 | 82.22 | 74.31 | 80.70 |
| BN (N-O) | 32.33 | 50.93 | 81.28 | 52.98 | 42.21 | 64.13 | 83.25 | 83.64 | 82.52 | 59.18 | 66.23 | 49.45 | 82.59 | 62.34 | 52.51 | 63.04 |
| TENT (N-O) | 31.39 | 40.27 | 75.68 | 42.03 | 35.38 | 64.32 | 84.92 | 84.96 | 81.43 | 46.84 | 49.48 | 39.77 | 84.21 | 49.23 | 43.49 | 56.89 |
| SHOT (N-O) | 30.69 | **37.69** | **61.97** | 41.30 | **34.74** | 54.19 | 76.33 | 71.94 | 74.24 | 46.50 | **47.98** | **38.88** | 70.60 | 46.09 | **40.74** | 51.59 |
| TTAC (N-O) | **30.36** | 38.84 | 69.06 | **39.67** | 36.01 | **50.20** | **66.18** | **70.17** | **64.36** | **45.59** | 51.77 | 39.72 | **62.43** | **44.56** | 42.80 | **50.11** |

normal domain shift in our life. The results in Table. 3 demonstrate our method is better able to adapt to the normal domain shift.

**TTT on Synthetic to Real Adaptation**. VisDA-C is a large-scale benchmark of synthetic-to-real object classification dataset. The setting of training on a synthetic dataset and testing on real data fits well with the real application scenario. On this dataset, we conduct experiments with our method under the N-O, Y-O and Y-M protocols and other methods under respective protocols, results are presented in Table. 5. We make the following observations. First, our method (TTAC Y-O) outperforms all methods except TTT++ under the Y-M protocol. This suggests TTAC is able to be deployed in the realistic test-time training protocol. Moreover, if training on the whole target data is allowed, TTAC (Y-M) further beats TTT++ by a large margin, suggesting the effectiveness of TTAC under a wide range of TTT protocols.

Table 3: Test-time training on CIFAR10.1.

| TEST | BN | TTT-R | TENT | SHOT | TTT++ | TTAC |
|---|---|---|---|---|---|---|
| 12.1 | 14.1 | 11.0 | 13.4 | 11.1 | 9.5 | **9.2** |

Table 4: Source-free sTTT on CIFAR10-C.

| TEST | BN | TENT | T3A | SHOT | TTAC | TTAC+SHOT |
|---|---|---|---|---|---|---|
| 29.15 | 15.49 | 14.27 | 15.44 | 13.95 | 13.74 | **13.35** |

## 4.5 Ablation Study

We conduct ablation study on CIFAR10-C dataset for individual components, including anchored clustering, pseudo label filtering, global feature alignment and finally the compatibility with contrastive branch [20]. For anchored clustering alone, we use all testing samples to update cluster statistics. For pseudo label filtering alone, we implement as predicting pseudo labels followed by filtering, then pseudo labels are used for self-training. We make the following observations from Tab. 6. Under both N-O and N-M protocols, introducing anchored clustering or pseudo label filtering alone improves over the baseline, e.g. under N-O $29.15\% \rightarrow 14.32\%$ for anchored clustering and $29.15\% \rightarrow 15.00\%$ for pseudo label filtering. When anchored clustering is combined with pseudo label filtering, we observe a significant boost in performance. This is due to more accurate estimation of category-wise cluster in the target domain and this reflects matching directly in the feature space may be better than minimizing cross-entropy with pseudo labels. We further evaluate aligning global features alone with KL-Divergence. This achieves relatively good performance and obviously outperforms the L2

Table 5: Test-time training on VisDA. The numbers for competing methods are inherited from [20].

| Method | Plane | Bcycl | Bus | Car | Horse | Knife | Mcycl | Person | Plant | Sktbrd | Train | Truck | Per-class |
|---|---|---|---|---|---|---|---|---|---|---|---|---|---|
| TEST | 56.52 | 88.71 | 62.77 | 30.56 | 81.88 | 99.03 | 17.53 | 95.85 | 51.66 | 77.86 | 20.44 | 99.51 | 65.19 |
| BN (N-M) [12] | 44.38 | 56.98 | 33.24 | 55.28 | 37.45 | 66.60 | 16.55 | 59.02 | 43.55 | 60.72 | 31.07 | 82.98 | 48.99 |
| TENT (N-M) [31] | 13.43 | 77.98 | 20.17 | 48.15 | 21.72 | 82.45 | 12.37 | 35.78 | 21.06 | 76.41 | 34.11 | 98.93 | 45.21 |
| SHOT (N-M) [19] | 5.73 | **13.64** | 23.33 | 42.69 | 7.93 | 86.99 | 19.17 | 19.97 | 11.63 | 11.09 | 15.06 | **43.26** | 25.04 |
| TFA (N-M) [20] | 28.25 | 32.03 | 33.67 | 64.77 | 20.49 | 56.63 | 22.52 | 36.30 | 24.84 | 35.20 | 25.31 | 64.24 | 37.02 |
| TTT++ (Y-M) [20] | 4.13 | 26.20 | 21.60 | **31.70** | 7.43 | 83.30 | 7.83 | 21.10 | 7.03 | **7.73** | **6.91** | 51.40 | 23.03 |
| TTAC (N-O) | 18.54 | 40.20 | 35.84 | 63.11 | 23.83 | 39.61 | 15.51 | 41.35 | 22.97 | 46.56 | 25.24 | 67.81 | 36.71 |
| TTAC (Y-O) | 7.19 | 29.99 | 22.52 | 56.58 | 8.14 | 18.41 | 8.25 | 22.28 | 10.18 | 23.98 | 13.55 | 67.02 | 24.01 |
| TTAC (Y-M) | **2.74** | 17.73 | **18.91** | 43.12 | **5.54** | **12.24** | **4.66** | **15.90** | **4.77** | 10.78 | 9.75 | 62.45 | **17.38** |

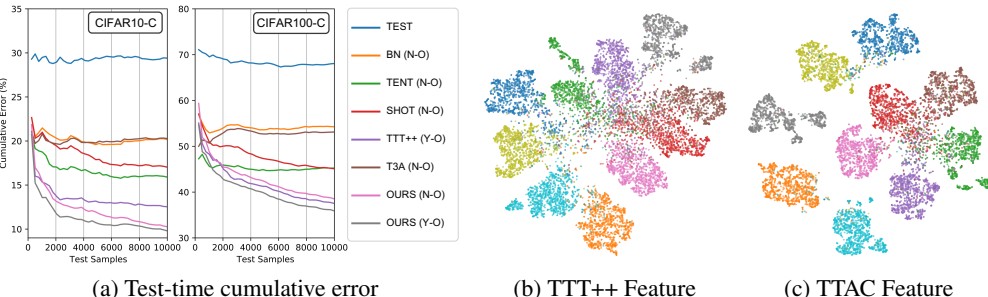

(a) Test-time cumulative error        (b) TTT++ Feature        (c) TTAC Feature

Figure 2: (a) Comparison of test-time cumulative error under one-pass protocol. (b) T-SNE visualization of TTT++ feature embedding. (c) T-SNE visualization of TTAC feature embedding.

distance alignment adopted in [20]. Finally, we combine all three components and the full model yields the best performance. When contrast learning branch is included, TTAC achieves even better results.

Table 6: Ablation study for individual components on CIFAR10-C dataset.

| TTT Protocol | - | N-O | | | | | Y-O | N-M | | | | | Y-M |
|---|---|---|---|---|---|---|---|---|---|---|---|---|---|
| Anchored Cluster. | - | ✓ | - | ✓ | - | ✓ | ✓ | ✓ | ✓ | - | - | ✓ | ✓ |
| Pseudo Label Filter. | - | - | ✓ | ✓ | - | ✓ | ✓ | - | ✓ | - | - | - | ✓ |
| Global Feat. Align. | - | - | - | - | KLD | KLD | KLD | - | - | L2 Dist.[20] | KLD | KLD | KLD |
| Contrast. Branch [20] | - | - | - | - | - | - | ✓ | - | - | - | - | - | ✓ |
| Avg Acc | 29.15 | 14.32 | 15.00 | 11.33 | 11.72 | 10.94 | 10.69 | 11.11 | 10.01 | 11.87 | 10.8 | 9.42 | 8.52 |

## 4.6 Additional Analysis

**Cumulative performance under sTTT**. We illustrate the cumulative error under the sTTT protocol in Fig. 2 (a). For both datasets TTAC outperforms competing methods from the early stage of test-time training. The advantage is consistent throughout the TTT procedure.

**TSNE Visualization of TTAC features**. We provide qualitative results for test-time training by visualizing the adapted features through T-SNE [30]. In Fig. 2 (b) and Fig. 2 (c), we compared the features learned by TTT++ [20] and TTAC (Ours). We observe a better separation between classes by TTAC, implying an improved classification accuracy.

**Source-Free Test-Time Training**. TTT aims to adapt model to target domain data by doing simultaneous training and sequential inference. It has been demonstrated some light-weight information, e.g. statistics, from source domain will greatly improve the efficacy of TTT. Nevertheless, under a more strict scenario where source domain information is strictly blind, TTAC can still exploit classifier prototypes to facilitate anchored clustering. Specifically, we normalize the category-wise weight vector with the norm of corresponding target domain cluster center as prototypes. Then, we build source domain mixture of Gaussians by taking prototypes as mean with a fixed covariance matrix. The results on CIFAR10-C are presented in Tab. 4. It is clear that even without any statistical information from source domain, TTAC still outperforms all competing methods.

**Test Sample Queue and Update Epochs.**    Under the sTTT protocol, we allow all competing methods to maintain the same test sample queue and multiple update epochs on the queue. To analyse the significance of the sample queue and update epochs, we evaluate BN, TENT, SHOT and TTAC on CIFAR10-C and ImageNet-C level 5 snow corruption evaluation set under different number of update epochs on test sample queue and under a without queue protocol, i.e. only update model w.r.t.

the current test sample batch. As the results presented in Tab. 7, we make the following observations. i) Maintaining a sample queue can substantially improve the performance of methods that estimate target distribution, e.g. TTAC ($11.91 \rightarrow 10.88$ on CIFAR10-C) and SHOT ($15.18 \rightarrow 13.96$ on CIFAR10-C). This is due to more test samples giving a better estimation of true distribution. ii) Consistent improvement can be observed with increasing update epochs for SHOT and TTAC. We ascribe this to iterative pseudo labeling benefiting from more update epochs.

Table 7: Comparing with and without test sample queue and different numbers of model update epochs. w/ Queue maintains a test sample queue with 4096 samples; w/o Queue maintains a single mini-batch with 256 and 128 samples on CIFAR10-C and ImageNet-C respectively.

| | CIFAR10-C | | | | | ImageNet-C | | |
| | w/ Queue | | | | w/o Queue | w/ Queue | | w/o Queue |
| #Epochs | 1 | 2 | 3 | 4* | 1 | 1 | 2* | 1 |
|---|---|---|---|---|---|---|---|---|
| BN | 15.84 | 15.99 | 16.04 | 16.00 | 15.44 | 62.34 | 62.34 | 62.59 |
| TENT | 13.35 | 13.83 | 13.85 | 13.87 | 13.48 | 47.82 | 49.23 | 48.39 |
| SHOT | 13.96 | 13.93 | 13.83 | 13.75 | 15.18 | 46.91 | 46.09 | 51.46 |
| TTAC | **10.88** | **10.80** | **10.58** | **9.96** | **11.91** | **45.44** | **44.56** | **46.64** |

**Computation Cost Measured in Wall-Clock Time.** Test sample queue and multiple update epochs introduce additional computation overhead. To investigate the impact on efficiency, we measure the overall wall time as the time elapsed from the beginning to the end of test-time training, including all I/O overheads. The per-sample wall time is then calculated as the overall wall time divided by the number of test samples. We report the per-sample wall time (in seconds) for BN, TENT, SHOT and TTAC in Tab. 8 under different update epoch settings and without queue setting. The Inference row indicates the per-sample wall time in a single forward pass including the data I/O overhead. We observe that, under the same experiment setting, BN and TENT are more computational efficient, but TTAC is only twice more expensive than BN and TENT if no test sample queue is preserved (0.0083 v.s. 0.0030/0.0041) while the performance of TTAC w/o queue is still better than TENT (11.91 v.s. 13.48). In summary, TTAC is able to strike a balance between computation efficiency and performance depending on how much computation resource is available. This suggests allocating a separate device is only necessary when securing best performance is the priority.

Table 8: The per-sample wall time (measured in seconds) on CIFAR10-C under sTTT protocol.

| | w/ Queue | | | | w/o Queue |
| #Epochs | 1 | 2 | 3 | 4 | 1 |
|---|---|---|---|---|---|
| BN | 0.0136 | 0.0220 | 0.0293 | 0.0362 | 0.0030 |
| TENT | 0.0269 | 0.0399 | 0.0537 | 0.0663 | 0.0041 |
| SHOT | 0.0479 | 0.0709 | 0.0942 | 0.1183 | 0.0067 |
| TTAC | 0.0516 | 0.0822 | 0.1233 | 0.1524 | 0.0083 |
| Inference | 0.0030 | 0.0030 | 0.0030 | 0.0030 | 0.0030 |

## 5 Conclusion

Test-time training (TTT) tackles the realistic challenges of deploying domain adaptation on-the-fly. In this work, we are first motivated by the confused evaluation protocols for TTT and propose two key criteria, namely modifying source training objective and sequential inference, to further categorize existing methods into four TTT protocols. Under the most realistic protocol, i.e. sequential test-time training (sTTT), we develop a test-time anchored clustering (TTAC) approach to align target domain features to the source ones. Unlike batchnorm and classifier prototype updates, anchored clustering allows all network parameters to be trainable, thus demonstrating stronger test-time training ability. We further propose pseudo label filtering and an iterative update method to improve anchored clustering and save memory footprint respectively. Experiments on six datasets verified the effectiveness of TTAC under sTTT as well as other TTT protocols.

**Acknowledgement** This work was supported in part by the National Natural Science Foundation of China (NSFC) under Grant 62106078, Guangdong RD key project of China (No.: 2019B010155001), and the Program for Guangdong Introducing Innovative and Enterpreneurial Teams (No.: 2017ZT07X183).

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
