# Appendix of "Revisiting Realistic Test-Time Training: Sequential Inference and Adaptation by Anchored Clustering"

**Yongyi Su**[1*]  **Xun Xu**[21*†]  **Kui Jia**[13†]

[1]South China University of Technology  [2]Institute for Infocomm Research
[3]Peng Cheng Laboratory
eesuyongyi@mail.scut.edu.cn
alex.xun.xu@gmail.com
kuijia@scut.edu.cn

In this appendix, we first provide more details for the derivation of iterative updating target domain cluster parameters. We further provide more details of the hyperparameters used in TTAC. Finally, we present evaluation of TTAC with transformer backbone, ViT [1], additional evaluation of TTAC update epochs, the stability of TTAC under different data streaming orders and compared alternative target clustering updating strategies.

## A   Derivations of Efficient Iterative Updating

The mean and covariance for each target domain cluster can be naively estimated through Maximum Likelihood Estimation (MLE) as below. The existing solution in TTT++ [4] stores the recent one thousand testing samples and their features for MLE.

$$\mu = \frac{1}{N} \sum_{i=1}^{N} f(x_i), \quad \Sigma = \frac{1}{N} \sum_{i=1}^{N} (f(x_i) - \mu)^\top (f(x_i) - \mu) \tag{1}$$

When $N$ is very large, it is inevitable that a very large memory space must be allocated to store all features $F \in \mathbb{R}^{N \times D}$, e.g. the VisDA dataset has 55k testing samples and a naive MLE prohibits efficient test-time training. In the manuscript, we propose to online update target domain feature distribution parameters without caching sample features as Eq. 8. The detailed derivations are now presented as follows. Formally, we denote the running mean and covariance at step $t - 1$ as $\mu^{t-1}$ and $\Sigma^{t-1}$, and the test minibatch at step $t$ as $\mathcal{B}^t = \{x_i\}_{i=1\cdots N_B}$. The following is the derivation of $\mu^t$.

$$\mu^t = \frac{1}{N^t} \sum_{i=1}^{N^t} f(x_i), \quad s.t. \quad N^t = N^{t-1} + |\mathcal{B}^t| \tag{2}$$

---

[*]Equal contribution
[†]Correspondence to <kuijia@scut.edu.cn> & <alex.xun.xu@gmail.com>

36th Conference on Neural Information Processing Systems (NeurIPS 2022).

$$\mu^t = \frac{1}{N^t}\left(\sum_{i=1}^{N^{t-1}} f(x_i) + \sum_{i=N^{t-1}+1}^{N^t} f(x_i)\right)$$

$$= \frac{1}{N^t}\left(N^{t-1}\cdot\mu^{t-1} + \sum_{i=N^{t-1}+1}^{N^t} f(x_i)\right) \tag{3}$$

$$= \mu^{t-1} + \frac{1}{N^t}\sum_{x_i\in\mathcal{B}^t}(f(x_i)-\mu^{t-1})$$

To simplify the expression, we denote $\delta^t = \frac{1}{N^t}\sum_{x_i\in\mathcal{B}^t}(f(x_i)-\mu^{t-1})$, so $\mu^t = \mu^{t-1} + \delta^t$. The following is the derivation of $\Sigma^t$. For the ease of calculation, we use the asymptotic unbiased estimator of $\Sigma^t$ as shown as below.

$$\Sigma^t = \frac{1}{N^t}\sum_{i=1}^{N^t}(f(x_i)-\mu^t)^\top(f(x_i)-\mu^t)$$

$$= \frac{1}{N^t}\sum_{i=1}^{N^t}(f(x_i)-\mu^{t-1}-\delta^t)^\top(f(x_i)-\mu^{t-1}-\delta^t)$$

$$= \frac{1}{N^t}\sum_{i=1}^{N^t}\{(f(x_i)-\mu^{t-1})^\top(f(x_i)-\mu^{t-1}) - \delta^{t\top}(f(x_i)-\mu^{t-1}) - (f(x_i)-\mu^{t-1})^\top\delta^t + \delta^{t\top}\delta^t\}$$

$$= \frac{1}{N^t}\left(\sum_{i=1}^{N^{t-1}}(f(x_i)-\mu^{t-1})^\top(f(x_i)-\mu^{t-1}) + \sum_{x_i\in\mathcal{B}^t}(f(x_i)-\mu^{t-1})^\top(f(x_i)-\mu^{t-1})\right.$$

$$\left. + \sum_{i=1}^{N^t}\{-\delta^{t\top}(f(x_i)-\mu^{t-1}) - (f(x_i)-\mu^{t-1})^\top\delta^t\}\right) + \delta^{t\top}\delta^t$$

$$= \frac{1}{N^t}\left(N^{t-1}\cdot\Sigma^{t-1} + \sum_{x_i\in\mathcal{B}^t}(f(x_i)-\mu^{t-1})^\top(f(x_i)-\mu^{t-1})\right) - \delta^{t\top}\delta^t$$

$$= \Sigma^{t-1} + \frac{1}{N^t}\sum_{x_i\in\mathcal{B}^t}\{(f(x_i)-\mu^{t-1})^\top(f(x_i)-\mu^{t-1}) - \Sigma^{t-1}\} - \delta^{t\top}\delta^t$$

$$\tag{4}$$

Furthermore, we give the formulations of the running mean $\mu_k^t$ and covariance $\Sigma_k^t$ for the $k^{th}$ target domain cluster as below.

$$\delta_k^t = \frac{1}{N_k^t}\sum_{x_i\in\mathcal{B}^t}F_i^{TC}F_i^{PP}\mathbb{1}(\hat{y}_i=k)(f(x_i)-\mu_k^{t-1}),$$

$$s.t.\quad N_k^t = N_k^{t-1} + \sum_{x_i\in\mathcal{B}^t}F_i^{TC}F_i^{PP}\mathbb{1}(\hat{y}_i=k)$$

$$\mu_k^t = \mu_k^{t-1} + \delta_k^t,$$

$$\Sigma_k^t = \Sigma_k^{t-1} + \frac{1}{N_k^t}\sum_{x_i\in\mathcal{B}^t}F_i^{TC}F_i^{PP}\mathbb{1}(\hat{y}_i=k)\{(f(x_i)-\mu_k^{t-1})^\top(f(x_i)-\mu_k^{t-1}) - \Sigma_k^{t-1}\} - \delta_k^{t\top}\delta_k^t$$

$$\tag{5}$$

Similarly to $N_{clip}$ for the threshold used to clip the $N^t$ protecting the gradient of new test samples, we use $N_{clip\_k}$ as the threshold to clip the $N_k^t$ for each target domain cluster.

Table 1: Hyper-parameters are used in our method.

| Dataset | $\alpha_k$ | $\beta_k$ | $N_C$ | $N_{itr}$ | $\xi$ | $\tau_{TC}$ | $\tau_{PP}$ | $N_{clip}$ | $N_{clip\_k}$ | $\lambda$ |
|---|---|---|---|---|---|---|---|---|---|---|
| CIFAR10-C | 0.1 | 0.1 | 4096 | 4 | 0.9 | -0.001 | 0.9 | 1280 | 128 | 1.0 |
| CIFAR100-C | 0.01 | 0.01 | 4096 | 4 | 0.9 | -0.001 | 0.9 | 1280 | 64 | 1.0 |
| CIFAR10.1 | 0.1 | 0.1 | 4096 | 4 | 0.9 | -0.001 | 0.9 | 1280 | 128 | 1.0 |
| VisDA-C | $\frac{1}{12}$ | $\frac{1}{12}$ | 4096 | 4 | 0.9 | -0.01 | 0.9 | 1536 | 128 | 1.0 |
| ModelNet40-C | 0.025 | 0.025 | 4096 | 6 | 0.9 | -0.1 | 0.5 | 1280 | 128 | 1.0 |
| ImageNet-C | 0.001 | 0.001 | 4096 | 2 | 0.9 | -0.01 | 0.9 | 1280 | 64 | 1.0 |

Table 2: The results using ViT backbone on CIFAR10-C dataset.

| Method | Bird | Contr | Defoc | Elast | Fog | Frost | Gauss | Glass | Impul | Jpeg | Motn | Pixel | Shot | Snow | Zoom | Avg | Std |
|---|---|---|---|---|---|---|---|---|---|---|---|---|---|---|---|---|---|
| TEST | 2.29 | 16.24 | 4.83 | 9.45 | 13.60 | 6.73 | 24.52 | 18.23 | 24.48 | 12.63 | 7.63 | 14.57 | 23.02 | 5.29 | 3.50 | 12.47 | 7.36 |
| BN | 2.29 | 16.24 | 4.83 | 9.45 | 13.60 | 6.73 | 24.52 | 18.23 | 24.48 | 12.63 | 7.63 | 14.57 | 23.02 | 5.29 | 3.50 | 12.47 | 7.36 |
| TENT | **1.84** | 3.55 | **3.31** | 7.01 | **5.57** | 4.09 | 60.97 | 10.20 | 61.12 | 9.72 | 4.93 | 3.87 | 22.47 | 4.55 | **2.64** | 13.72 | 19.19 |
| SHOT | 2.00 | **3.13** | 3.46 | 6.63 | 5.79 | 4.06 | 11.65 | 9.39 | 10.58 | 9.69 | 5.03 | 3.63 | 10.05 | 4.35 | 2.70 | 6.14 | 3.15 |
| TTT++ | 1.91 | 4.14 | 3.88 | **6.58** | 6.27 | 4.00 | 10.08 | 8.59 | 8.85 | 9.66 | **4.68** | **3.62** | 9.17 | 4.28 | 2.74 | 5.90 | 2.64 |
| TTAC (Ours) | 2.15 | 4.05 | 3.91 | 6.62 | 5.67 | **3.75** | **9.26** | **7.95** | **7.97** | **8.55** | 4.75 | 3.87 | **8.24** | **3.93** | 2.94 | **5.57** | **2.24** |

# B  Hyperparameter Values

We provide the details of hyperparameters in this section. Hyperparameters are shared across multiple TTT protocols except for $N_C$ and $N_{itr}$ which are only applicable under one-pass adaptation protocols. The details are shown as Tab. 1. $\alpha_k$ and $\beta_k$ respectively represent the prevalence of each category, here we set them to 1 over the number of categories. $N_C$ indicates the length of the testing sample queue $C$ under the sTTT protocol, and $N_{itr}$ controls the update epochs on this queue. $\tau_{TC}$ and $\tau_{PP}$ are the thresholds used for pseudo label filtering. $N_{clip}$ and $N_{clip\_k}$ are the upper bounds of sample counts in the iterative updating of global statistics and target cluster statistics respectively. Finally $\lambda$ is the coefficient of $\mathcal{L}_{ga}$, which takes the default value of 1. All models are implemented by the PyTorch 1.10.2 framework, CUDA 11.3 with an NVIDIA RTX 3090 GPU.

# C  Additional Evaluation

## C.1  Evaluation of TTAC with Transformer Backbone

In this section, we provide additional evaluation of TTAC with a transformer backbone, ViT [1]. In specific, we pre-train ViT on CIFAR10 clean dataset and then follow the sTTT protocol to do test-time training on CIFAR10-C. The results are presented in Tab. 2. We report the average (Avg) and standard deviation (Std) of accuracy over all 15 categories of corruptions. Again, TTAC consistently outperform all competing methods with transformer backbone.

## C.2  Impact of TTAC Update Epochs on Cached Testing Sample

Under the sTTT protocol, we perform multiple iterations of adaptation on cached testing sample queue. Preserving a history of testing samples is a commonly practice in test-time training. For example, T3A [2] preserves a support set, which contains testing samples and the pseudo labels, to update classifier prototypes. TTT++ [4] preserves a testing sample queue to estimate global feature distribution. For these methods, both raw testing samples and features must be cached simultaneously, in comparison, we only cache the raw data samples and target domain clusters are estimated in an online fashion.

Here, we analyze the impact of TTAC update epochs on cached testing samples. The results are presented in Tab. 3, where we make the following observations. First, the error rate is decreasing as the number of epochs increases, while at the cost of more computation time. But this can be solved by allocating a separate device for model adaptation. Second, the error rate saturates at $N_{itr} = 4$ suggesting only a few epochs is necessary to achieve good test-time training on target domain.

Table 3: The impact of TTAC update epochs under the sTTT protocol.

| $N_{itr}$ | Bird | Contr | Defoc | Elast | Fog | Frost | Gauss | Glass | Impul | Jpeg | Motn | Pixel | Shot | Snow | Zoom | Avg |
|---|---|---|---|---|---|---|---|---|---|---|---|---|---|---|---|---|
| 1 | 6.57 | 8.20 | 8.57 | 15.82 | 11.61 | 11.60 | 17.46 | 22.66 | 20.99 | 11.97 | 10.44 | 13.79 | 15.40 | 10.96 | 7.49 | 12.90 |
| 2 | 6.82 | 8.12 | 8.77 | 15.96 | 11.79 | 11.17 | 15.49 | 23.53 | 19.78 | 12.28 | 10.19 | 13.22 | 16.28 | 10.84 | 7.49 | 12.78 |
| 3 | 6.80 | 8.11 | 8.53 | 15.94 | 11.36 | 10.89 | 14.87 | 22.67 | 18.94 | 11.77 | 9.83 | 12.51 | 15.91 | 10.58 | 7.35 | 12.40 |
| **4** | **6.41** | 8.05 | **7.85** | 14.81 | **10.28** | **10.51** | **13.06** | 18.36 | **17.35** | **10.80** | 8.97 | **9.34** | **11.61** | **10.01** | **6.68** | **10.94** |
| 6 | 6.42 | **7.64** | 7.97 | **14.66** | 10.66 | 10.59 | 13.30 | **18.29** | 17.61 | 10.86 | **8.94** | 9.36 | 11.76 | 10.03 | 6.73 | 10.98 |

Table 4: The performance of TTAC under different data streaming orders.

| Random Seed | 0 | 10 | 20 | 200 | 300 | 3000 | 4000 | 40000 | 50000 | 500000 | Avg |
|---|---|---|---|---|---|---|---|---|---|---|---|
| Error (%) | 10.01 | 10.06 | 10.05 | 10.29 | 10.20 | 10.03 | 10.31 | 10.36 | 10.37 | 10.13 | 10.18±0.13 |

## C.3 Impact of Data Streaming Order

The proposed sTTT protocols assumes test samples arrive in a stream and inference is made instantly on each test sample. The result for each test sample will not be affected by any following ones. In this section, we investigate how the data streaming order will affect the results. Specifically, we randomly shuffle all testing samples in CIFAR10-C for 10 times with different seeds and calculate the mean and standard deviation of test accuracy under sTTT protocol. The results in Tab. 4 suggest TTAC maintains consistent performance regardless of data streaming order.

## C.4 Alternative Strategies for Updating Target Domain Clusters

In the manuscript, we presented target domain clustering through pseudo labeling. A temporal consistency approach is adopted to filter out confident samples to update target clusters. In this section, we discuss two alternative strategies for updating target domain clusters. Firstly, each target cluster can be updated with all samples assigned with respective pseudo label (Without Filtering). This strategy will introduce many noisy samples into cluster updating and potentially harm test-time feature learning. Secondly, we use a soft assignment of testing samples to each target cluster to update target clusters (Soft Assignment). This strategy is equivalent to fitting a mixture of Gaussian through EM algorithm. Finally, we compare these two alternative strategies with our temporal consistency based filtering approach. The results are presented in Tab. 5. We find the results with temporal consistency based filtering outperforms the other two strategies on 13 out of 15 categories of corruptions, suggesting pseudo label filtering is necessary for estimating more accurate target clusters.

## C.5 Sensitivity to Hyperparameters

We evaluate the sensitivity to two thresholds during pseudo label filtering, namely the temporal smoothness threshold $\tau_{TC}$ and posterior threshold $\tau_{PP}$. $\tau_{TC}$ controls how much the maximal probability deviate from the historical exponential moving average. If the current value is lower than the ema below a threshold, we believe the prediction is not confident and the sample should be excluded from estimating target domain cluster. $\tau_{PP}$ controls the the minimal maximal probability and below this threshold is considered as not confident enough. We evaluate $\tau_{TC}$ in the interval between 0 and -1.0 and $\tau_{PP}$ in the interval from 0.5 to 0.95 with results on CIFAR10-C level 5 glass blur corruption presented in Tab. 6. We draw the following conclusions on the evaluations. i) There is a wide range of hyperparameters that give stable performance, e.g. $\tau_{TC} \in [0.5, 0.0.9]$ and $\tau_{PP} \in [-0.0001, -0.01]$. ii) When temporal consistency filtering is turn off, i.e. $\tau_{TC} = -1.0$, because the probability is normalized to between 0 and 1, the performance drops substantially, suggesting the necessity to apply temporal consistency filtering.

Table 5: Comparison of alternative strategies for updating target domain clusters.

| Strategy | Bird | Contr | Defoc | Elast | Fog | Frost | Gauss | Glass | Impul | Jpeg | Motn | Pixel | Shot | Snow | Zoom | Avg |
|---|---|---|---|---|---|---|---|---|---|---|---|---|---|---|---|---|
| i. Without filtering | 7.19 | 8.98 | 9.29 | 17.28 | 11.90 | 11.72 | 17.19 | 22.47 | 20.83 | 12.27 | 10.11 | 12.39 | 13.85 | 11.56 | 7.97 | 13.00 |
| ii. Soft Assignment | 6.77 | **8.02** | 7.93 | **14.77** | 10.87 | 10.68 | 13.65 | 18.69 | 17.58 | 11.26 | 9.33 | 9.54 | 11.70 | 10.56 | 6.93 | 11.22 |
| Filtering (Ours) | **6.41** | 8.05 | **7.85** | 14.81 | **10.28** | **10.51** | **13.06** | **18.36** | **17.35** | **10.80** | **8.97** | **9.34** | **11.61** | **10.01** | **6.68** | **10.94** |

Table 6: Evaluation of pseudo labeling thresholds on CIFAR10-C level 5 glass blur corruption. Numbers are reported as classification error (%).

| $\tau_{TC}\backslash\tau_{PP}$ | 0.5 | 0.6 | 0.7 | 0.8 | 0.9 | 0.95 |
|---|---|---|---|---|---|---|
| 0.0 | 23.03 | 22.26 | 21.96 | 22.50 | 21.14 | 28.55 |
| -0.0001 | 20.03 | 20.53 | 20.45 | 20.40 | 19.49 | 27.00 |
| -0.001 | 19.66 | 20.51 | 19.49 | 20.48 | **19.42** | 26.83 |
| -0.01 | 20.71 | 20.78 | 20.73 | 20.65 | 20.29 | 27.58 |
| -0.1 | 24.10 | 21.47 | 21.46 | 22.36 | 21.45 | 28.71 |
| -1.0 | 30.75 | 24.08 | 23.40 | 24.33 | 22.21 | 28.77 |

**C.6 Improvement by KL-Divergence**

Minimizing KL-Divergence between two Gaussian distributions is equivalent to matching the first two moments of the true distributions [3]. TFA or TTT++ aligns the first two moments through minimizing the L2/F norm, referred to as L2 alignment hereafter. Although L2 alignment is derived from Central Moment Discrepancy [5], the original CMD advocates a higher order moment matching and the weight applied to each moment is hard to estimate on real-world datasets. An empirical weight could be applied to balance the mean and covariance terms in TTT++, at the cost of introducing additional hyperparameters. We also provide a comparison between KL-Divergence and L2 alignment on CIFAR10-C level 5 snow corruption in Tab. 7 using the original code released by TTT++. The performance gap empirically demonstrates the superiority of KL-Divergence. Nevertheless, we believe a theoretical analysis into why KL-Divergence is superior under test-time training would be inspirational and we leave it for future work.

Table 7: Comparing KL-Divergence and L2 alignment as test-time training loss with the original code released by TTT++ (Y-M) on CIFAR10 level 5 snow corruption.

| Feature Alignment Strategy | Error (%) |
|---|---|
| L2 alignment (original TTT++) | 9.85 |
| KL-Divergence | 8.43 |

# D Limitations and Failure Cases

We discuss the limitations of our method from two perspectives. First, we point out that TTAC implements backpropagation to update models at test stage, therefore additional computation overhead is required. Specifically, as Tab. **??**, we carried out additional evaluations on the per-sample wall clock time. Basically, we discovered that TTAC is 2-5 times computationally more expensive than BN and TENT. However, contrary to usual recognition, BN and TENT are also very expensive compared with no adaptation at all. Eventually, most test-time training methods might require an additional device for test-time adaptation.

We further discuss the limitations on test-time training under more severe corruptions. Specifically, we evaluate TENT, SHOT and TTAC under 1-5 levels of corruptions on CIFAR10-C with results reported in Tab. 8. We observe generally a drop of performance from 1-5 level of corruption. Despite consistently outperforming TENT and SHOT at all levels of corruptions, TTAC's performance at higher corruption levels are relatively worse, suggesting more attention must be paid to more severely corrupted scenarios.

# E Detailed results

We further provide details of test-time training on CIFAR10-C, CIFAR100-C and ModelNet40-C datasets in Tab. 9, 10 and 11 respectively. The results in Tab. 9 and 10 suggest TTAC has a powerful ability to adapt to the corrupted images, and obtains the state-of-the-art performances on almost all corruption categories.

Table 8: Classification error under different levels of snow corruption on CIFAR10-C dataset.

| Level | 1 | 2 | 3 | 4 | 5 |
|---|---|---|---|---|---|
| TEST | 9.46 | 18.34 | 16.89 | 19.31 | 21.93 |
| TENT | 8.76 | 11.39 | 13.37 | 15.18 | 13.93 |
| SHOT | 8.70 | 11.21 | 13.16 | 15.12 | 13.76 |
| TTAC | 6.54 | 8.19 | 9.82 | 10.61 | 9.98 |

Table 9: The results of CIFAR10-C under the sTTT protocol

| Method | Bird | Contr | Defoc | Elast | Fog | Frost | Gauss | Glass | Impul | Jpeg | Motn | Pixel | Shot | Snow | Zoom | Avg |
|---|---|---|---|---|---|---|---|---|---|---|---|---|---|---|---|---|
| TEST | 7.00 | 13.28 | 11.84 | 23.38 | 29.42 | 28.25 | 48.73 | 50.79 | 57.01 | 19.46 | 23.38 | 47.88 | 44.00 | 21.93 | 10.84 | 29.15 |
| BN | 8.21 | 8.36 | 9.73 | 19.43 | 20.16 | 13.72 | 17.46 | 26.34 | 28.11 | 14.00 | 13.90 | 12.22 | 16.64 | 16.00 | 8.03 | 15.49 |
| TENT | 8.22 | 8.07 | 9.93 | 18.29 | 15.65 | 14.14 | 16.60 | 24.10 | 25.80 | 13.39 | 12.34 | 11.06 | 14.75 | 13.87 | 7.87 | 14.27 |
| T3A | 8.33 | 8.70 | 9.70 | 19.51 | 20.26 | 13.83 | 17.27 | 25.61 | 27.63 | 14.05 | 14.26 | 12.12 | 16.37 | 15.78 | 8.13 | 15.44 |
| SHOT | 7.58 | 7.78 | 9.12 | 17.76 | 16.90 | 12.56 | 15.99 | 23.30 | 24.99 | 13.19 | 12.59 | 11.37 | 14.85 | 13.75 | 7.51 | 13.95 |
| TTT++ | 7.70 | 7.91 | 9.24 | 17.55 | 16.39 | 12.74 | 15.49 | 22.57 | 22.86 | 13.02 | 12.52 | 11.46 | 14.45 | 13.90 | 7.51 | 13.69 |
| TTAC (Ours) | 6.41 | 8.05 | 7.85 | 14.81 | **10.28** | **10.51** | **13.06** | 18.36 | **17.35** | **10.80** | 8.97 | 9.34 | **11.61** | **10.01** | **6.68** | **10.94** |
| TTAC+SHOT (Ours) | **6.37** | **6.98** | **7.79** | **14.80** | 11.04 | 10.52 | 13.58 | **18.34** | 17.68 | 10.94 | **8.93** | **9.20** | 11.81 | 10.01 | 6.79 | 10.99 |

Table 10: The results of CIFAR100-C under the sTTT protocol

| Method | Bird | Contr | Defoc | Elast | Fog | Frost | Gauss | Glass | Impul | Jpeg | Motn | Pixel | Shot | Snow | Zoom | Avg |
|---|---|---|---|---|---|---|---|---|---|---|---|---|---|---|---|---|
| TEST | 28.84 | 50.87 | 39.61 | 59.53 | 68.10 | 60.21 | 80.77 | 82.27 | 87.75 | 49.98 | 54.20 | 72.27 | 77.84 | 54.57 | 38.36 | 60.34 |
| BN | 31.78 | 33.06 | 33.86 | 48.65 | 54.23 | 42.28 | 48.02 | 57.08 | 60.14 | 39.09 | 40.72 | 37.76 | 45.83 | 46.31 | 31.91 | 43.38 |
| TENT | 30.45 | 31.47 | 32.48 | 45.84 | 44.85 | 41.39 | 45.59 | 52.31 | 56.16 | 38.94 | 38.41 | 35.55 | 43.40 | 42.89 | 31.10 | 40.72 |
| T3A | 31.66 | 32.63 | 33.62 | 47.60 | 53.06 | 41.95 | 46.63 | 55.51 | 58.92 | 38.89 | 40.26 | 37.21 | 45.32 | 46.08 | 31.43 | 42.72 |
| SHOT | 29.36 | 30.49 | 31.33 | 43.41 | 45.14 | 39.31 | 43.35 | 50.98 | 53.75 | 36.07 | 36.11 | 34.54 | 42.16 | 40.99 | 29.52 | 39.10 |
| TTT++ | 30.79 | 31.48 | 33.04 | 44.95 | 47.74 | 40.19 | 43.94 | 52.06 | 54.08 | 37.26 | 38.10 | 35.40 | 42.28 | 42.97 | 30.58 | 40.32 |
| TTAC (Ours) | 28.13 | 32.55 | 29.45 | 41.54 | 39.07 | 36.95 | 40.01 | 48.30 | 49.21 | 34.55 | 33.29 | 32.69 | 38.62 | 37.69 | 27.61 | 36.64 |
| TTAC+SHOT (Ours) | **27.73** | 32.19 | **29.25** | **41.26** | **38.67** | **36.67** | 40.01 | **47.87** | 49.21 | **34.13** | **32.98** | **32.52** | 38.62 | **37.35** | **27.36** | **36.39** |

Table 11: The results of ModelNet40-C under the sTTT protocol

| Method | Background | Cutout | Density Inc. | Density Dec. | Inv. RBF | RBF | FFD | Gaussian | Impulse | LiDAR | Occlusion | Rotation | Shear | Uniform | Upsampling | Avg |
|---|---|---|---|---|---|---|---|---|---|---|---|---|---|---|---|---|
| TEST | 57.41 | 23.82 | 16.17 | 27.59 | 21.19 | 22.85 | 19.89 | 27.07 | 37.48 | 85.21 | 65.24 | 41.61 | 16.33 | 22.93 | 34.44 | 34.62 |
| BN | 52.88 | 18.07 | 13.25 | 20.42 | 16.57 | 17.50 | 17.75 | 17.30 | 18.60 | 70.75 | 58.51 | 26.94 | 14.51 | 15.48 | 19.37 | 26.53 |
| TENT | 51.94 | 17.38 | 13.25 | 17.99 | 14.14 | 16.65 | 15.68 | 16.49 | 17.10 | 81.44 | 64.18 | 22.33 | 13.29 | 14.59 | 19.25 | 26.38 |
| T3A | 52.51 | 16.37 | 13.09 | 18.23 | 14.26 | 15.48 | 15.88 | 14.14 | 15.68 | 69.12 | 54.82 | 24.80 | 13.01 | 14.14 | 17.06 | 24.57 |
| SHOT | **15.64** | **14.34** | 12.24 | **15.48** | 13.37 | **13.82** | **12.64** | 13.13 | **13.43** | 66.05 | 47.41 | **18.80** | **11.79** | 12.44 | 15.11 | 19.71 |
| TTAC (Ours) | 24.88 | 17.14 | 12.44 | 19.12 | 15.07 | 16.29 | 16.45 | 14.95 | 16.37 | 63.49 | 52.19 | 22.41 | 13.70 | 13.78 | 16.21 | 22.30 |
| TTAC+SHOT (Ours) | 18.67 | 14.89 | **10.88** | 15.58 | **13.12** | 14.19 | 14.04 | **12.15** | 14.08 | **57.35** | 47.48 | 18.93 | 11.99 | **11.92** | **12.88** | **19.21** |