# OpenReview forum: "Revisiting Realistic Test-Time Training: Sequential Inference and Adaptation by Anchored Clustering"
_NeurIPS.cc/2022/Conference — NeurIPS 2022 Accept_

### Official Review · Reviewer_6icX · 2022-07-11

**Rating:** 6
**Confidence:** 5
**Soundness:** 2 fair
**Presentation:** 3 good
**Contribution:** 3 good

**Summary:**

Test-time training is an emerging approach for building robust machine learning models to distribute shifts. The main contribution of this paper summarizes the following two points. (1) They distinguish experiment setups, from the perspective of whether it uses on-pass adaptation or multi-pass adaptation and whether it needs to alter the training phase.  The distinction is important as prior works often compared different setups, leading to unfair comparisons. (2)  They provide a new method on realistic test-time training setup, called test-time anchored clustering. Specifically, they use a mixture of Gaussians to model the clusters in the target domain, and use pseudo-label to match it with the mixture distribution of source data in closed form. Empirical results show that the proposed method outperforms existing methods on several benchmark datasets.


**Questions:**

(1) Why do we need to represent each distribution by a mixture of Gaussian distributions? In other words, is it better than simply aligning the statistics (e.g, mean and covariance) of source data of a category and that of the target data predicted as the category?

(2) How did you tune the hyper parameters for test-time training methods?

(3) Why is SHOT categorized as N-O or N-M? In my understanding, the original SHOT contains several proposals including label-smoothing techniques for source training, and thus require special objective function for source training. Does the value reported here not use label-smoothing?

(4) According to Table 5, major performance gains come from global feature alignment using KLD. In my understanding, several prior methods (e.g., TFA and TTT+) use similar feature alignment loss to match the statistics of source and target distribution. Does the results suggest KLD is better than other feature alignment approaches, or does there any difference between setup among these methods?


**Limitations:**

This paper does not provide limitation section in the main text. I recommend the author to discuss the potential limitation of the proposed method (e.g. hyper parameter tuning mentioned in weakness in my review), and potential drawback of test-time training itself (e.g., do we really want to adapt the model during test-time in the safety critical application?).

**Strengths And Weaknesses:**

---
**Strengths**

(1) The paper is well written and easy to follow.

(2)  They pointed out important confusion about evaluation in the fields of test-time training. I strongly agree that the confusion often leads to unfair comparisons, and the clarification of this paper makes it easier for subsequent research to conduct fairer comparisons, which I think very important for scientific progress.

(3) This paper provides extensive empirical validations, carefully investigating how each component of the proposal contributes to the performance gain.

---
**Weaknesses**

(1) Why do we need to represent each distribution by a mixture of Gaussian distributions? In other words, is it better than simply aligning the statistics (e.g, mean and covariance) of source data of a category and that of the target data predicted as the category? The former approach has been already proposed in test-time adaptation recently [1], it is better to clarify the difference.

[1] Kojima, Takeshi et al. “Robustifying Vision Transformer without Retraining from Scratch by Test-Time Class-Conditional Feature Alignment.” IJCAI2022

(2) Comparing sTTA and other TTA setup, one notable difficulty of sTTA might be how to choose the hyperparameter. Because there is no clue about test-distribution when we choose the hyper parameter (e.g., learning rate, optimizer, tau for the threshold, etc) in sTTA, there is a reasonable way to determine the hyper parameter. From the current manuscript, unfortunately, how to tune the hyperparameter for each test-time training method is unclear for me, which makes me wonder whether the reported results are actually fair or not.

(3) According to Table 5, major performance gains come from global feature alignment using KLD. In my understanding, several prior methods (e.g., TFA and TTT+) use similar feature alignment loss to match the statistics of source and target distribution. Does the results suggest KLD is better than other feature alignment approaches, or does there any difference between setup among these methods?

(Minor comments)
- line 124 needs a period.
- line 131: should clarify what is u and sigma (d-dimensional mean vector and covariance matrix?)

---

> ### Author Response · Authors · 2022-08-02
> **Response to Reviewer 6icX**
>
>
> We would like thank the reviewer for acknowledging the contributions in categorizing TTT protocols and recommendations to explain hyperparameter tuning. In the response, we shall explain hyperparameter tuning and provide clarifications on other concerns. We hope these could ease the concerns and earn support to this work.
>
> ### Represent Distribution by Mixture of Gaussians
>
> We consider using a mixture of Gaussians (MoG) to model of target data distribution because the feature distribution is obviously multi-modal. In practice, considering that no-closed form solution exists for the KL-Divergence between two MoGs and the feature distribution within a semantic class is more likely to be modelled by a single Gaussian, we choose to align each component Gaussian between the source and target domains and each component Gaussian corresponds to one category. In the target domain each component Gaussian is estimated by the predicted samples. Directly aligning the MoGs or more sophisticated distributions deserves investigation as future work.
>
> We also appreciate the work, Kojima et al. IJCAI2022 (CFA), suggested by the reviewer. After carefully reviewing this work, we summarize two key differences. First, CFA does not filter out any pseudo labels, which are often highly noisy. We argue that this filtering is necessary, as shown in our ablation study removing pseudo label filtering results in 3% increase in error rate ($11.33\%\rightarrow14.32\%$) under the N-O protocol. Moreover, CFA aligns the first order raw moment (mean) and in our experiments we discover that this is suboptimal compared to KL-Divergence in the sTTT protocol. In the revised manuscript, we shall discuss relations to this work from the above two perspectives.
>
> ### Hyperparemter Tuning
>
> We acknowledge that automatic hyperparameter tuning under the TTT protocol is non-trivial since holding out a separate validation requires knowing the target domain in advance which violates the TTT assumption. In practice, we follow the approach proposed in TTT++ by holding out a 10% subset of each corruption data to form a mixed corruption set and estimate the hyperparameters, including learning rate and thresholds. The optimizer is empirically chosen to be SGD. For competing methods, we always use the hyperparameters released in respective code repository. A more detailed analysis into the sensitivity to pseudo labeling thresholds can be found in the response to reviewer Gh1A.
>
> ### Categorization of SHOT
>
> Thanks very much for the remind. We made two changes to SHOT for fair comparison with other methods. First, we presented SHOT under N-O by only allowing sequential adaptation, i.e. using past and current batch for update. Second, we presented SHOT under N-M by initializing test-time training from a pretrained model which is shared by all competing methods under N-M/O protocols. Since we advocate the N-O protocol, SHOT under Y-M/O is not evaluated at this moment. We shall explicitly discuss the practice and reason for implementing SHOT under N-M/O protocols in the revised manuscript.
>
>
> ### Improvement by KL-Divergence
>
> Minimizing KL-Divergence between two Gaussian distributions is equivalent to matching the first two moments of the true distributions [1]. TFA or TTT++ aligns the first two moments through minimizing the L2/F norm, referred to as L2 alignment hereafter. Although L2 alignment is derived from Central Moment Discrepancy [2], the original CMD advocates a higher order moment matching and the weight applied to each moment is hard to estimate on real-world datasets. An empirical weight could be applied to balance the mean and covariance terms in TTT++, at the cost of introducing additional hyperparameters. We also provide a comparison between KL-Divergence and L2 alignment on CIFAR10-C level 5 snow corruption in Table 1 using the original code released by TTT++. The performance gap empirically demonstrates the superiority of KL-Divergence. We shall discuss the comparison in the revised manuscript and point out that a theoretical analysis into why KL-Divergence is superior under test-time training could be the future work.
>
> [1] Kurz G, Pfaff F, Hanebeck U D. Kullback-leibler divergence and moment matching for hyperspherical probability distributions[C]//2016 19th International Conference on Information Fusion (FUSION). IEEE, 2016: 2087-2094.
>
> [2] Zellinger W, Grubinger T, Lughofer E, et al. Central moment discrepancy (cmd) for domain-invariant representation learning[J]. arXiv preprint arXiv:1702.08811, 2017.
>
>
>
> Table 1: Comparing KL-Divergence and L2 alignment as test-time training loss with the original code released by TTT++ (Y-M) on CIFAR10  level 5 snow corruption.
>
> |  Feature Alignment Strategy  | Error (%) |
> |:----------------------------:|:---------:|
> | L2 distance (original TTT++) |    9.85   |
> |              KLD             |  **8.43** |

---

> > ### Comment · Reviewer_6icX · 2022-08-08
> > **Confirm my score**
> >
> > Thanks for your feedback. The rebuttal clarify my concerns, and would like to keep my current score.

---

> > > ### Author Response · Authors · 2022-08-09
> > > **Response to Reviewer 6icX's Additional Comment**
> > >
> > > We would like to thank the reviewer for supporting this work. In the revised manuscript, we shall explicitly address the hyperparameter tuning, categorization of SHOT and point out that more theoretical analysis into KL-Divergence as alignment objective is necessary as future work. Thanks again for your firm support.

---

### Official Review · Reviewer_pHz1 · 2022-07-11

**Rating:** 7
**Confidence:** 4
**Soundness:** 3 good
**Presentation:** 3 good
**Contribution:** 3 good

**Summary:**

In spite of the numerous efforts put into test-time training (TTT), the experimental settings are unclear, resulting in unfair comparisons.
This paper revisits TTT assumptions and classifies TTT protocols. This paper aims to tackle on the most realistic and challenging TTT protocol (one-pass test time training with no modifications to training objective). Motivated by the confused evaluation protocols for TTT, the authors propose test-time anchored clustering (TTAC) approach to align target domain features to the source ones in sequential test-time training. On five TTT datasets, TTAC consistently beats state-of-the-art approaches.

**Questions:**

- Sensitivity analysis for key hyperparameters such as thresholds used for pseudo label filtering related was not presented. I would like to know the trends related to key hyperparameters.

**Ethics Review Area:**

["I don’t know"]

**Limitations:**

There is no description of the limitations or potential negative social impacts of the proposed method.

**Strengths And Weaknesses:**

Strength:
- This paper is well-written and the whole part including motivation, method and experiment are easy to understand.
- Clear and detailed descriptions of the proposed method, which provide enough information to reproduce.
- This paper presents a categorization of TTT protocols based on two key factors so that comparisons of TTT methods within each category could be fair.
- This paper shows the experimental results for various experimental protocols for each method, which can serve as a baseline for subsequent TTT-related studies.

Weakness:
- This paper is not the first attempt to use statistical information between the source domain and the target domain to alleviate domain discrepancy. [1] was not mentioned even though it is very relevant to this paper in that it reduces the difference between the centroid (mean) of the source domain and the centroid (mean) of the target domain via pseudo labels or uses a moving average scheme. Although this paper uses covariance in addition to mean, novelty is not enough.
- On line 125, there must be a space between alone and we.


[1] Xie, Shaoan, et al. "Learning semantic representations for unsupervised domain adaptation." ICML 2018.

---

> ### Author Response · Authors · 2022-08-02
> **Response to Reviewer pHz1**
>
>
> We highly appreciate the invaluable comments and suggestions. In the response, we addressed the sensitivity to hyperparameters and discuss the concerns over novelty. We hope the response could ease the concerns from the reviewer and earn firm support to this work.
>
>
> ### Novelty Concerns
>
> We thank the reviewer for referring to Xie et al., ICML 2018. After carefully reviewing the recommended paper, we summarize the following differences. In Xie et al. ICML 2018, aligning source and target domain centers by minimizing the L2 distance is proposed. However, as we empirically discovered in our submission, aligning the distribution with KL-Divergence is more effective as it considers the high order statistics. Moreover, we introduced a test sample queue to allow better distribution fitting and developed an incremental update mechanism to save memory footprint.
>
>
> ### Sensitivity to Hyperparameters
>
> We evaluate the sensitivity to two thresholds during pseudo label filtering, namely the temporal smoothness threshold $\tau_{TC}$ and posterior threshold $\tau_{PP}$. $\tau_{TC}$ controls how much the maximal probability deviate from the historical exponential moving average. If the current value is lower than the ema below a threshold, we believe the prediction is not confident and the sample should be excluded from estimating target domain cluster. $\tau_{PP}$ controls the the minimal maximal probability and below this threshold is considered as not confident enough. We evaluate $\tau_{TC}$ in the interval between 0 and -1.0 and $\tau_{PP}$ in the interval from 0.5 to 0.95 with results on CIFAR10-C level 5 glass blur corruption presented in Table 1. We draw the following conclusions on the evaluations. i) There is a wide range of hyperparameters that give stable performance, e.g. $\tau_{TC}\in[0.5,0.0.9]$ and $\tau_{PP}\in[-0.01, -0.0001]$. ii) When temporal consistency filtering is turn off, i.e. $\tau_{TC}=-1.0$, because the probability is normalized to between 0 and 1, the performance drops substantially, suggesting the necessity to apply temporal consistency filtering.
>
> Table 1: Evaluation of pseudo labeling thresholds on CIFAR10-C level 5 glass blur corruption. Numbers are reported as classification error (%).
>
> | $\tau_{TC}\backslash \tau_{PP}$ |  0.5  |  0.6  |   0.7  |  0.8  |  0.9  |  0.95 |
> |:-------|:-----:|:-----:|:------:|:-----:|:-----:|:-----:|
> | **0.0**     | 23.03 | 22.26 |  21.96 | 22.50 | 21.14 | 28.55 |
> | **-0.0001** | 20.03 | 20.53 |  20.45 | 20.40 | 19.49 | 27.00 |
> | **-0.001**  | 19.66 | 20.51 |  19.49 | 20.48 | 19.42 | 26.83 |
> | **-0.01**   | 20.71 | 20.78 |  20.73 | 20.65 | 20.29 | 27.58 |
> | **-0.1**    | 24.10 | 21.47 |  21.46 | 22.36 | 21.45 | 28.71 |
> | **-1.0**    | 30.75 | 24.08 |  23.40 | 24.33 | 22.21 | 28.77 |

---

### Official Review · Reviewer_Gh1A · 2022-07-11

**Rating:** 6
**Confidence:** 4
**Soundness:** 3 good
**Presentation:** 3 good
**Contribution:** 3 good

**Summary:**

The paper proposes a clustering-based method for sequential test-time training, where the test data under distribution shifts arrives in a stream manner. The core idea of the proposed method is to match the cluster statistics between the source and target domain. To obtain a robust estimate of the cluster statistics, the paper introduces a pseudo label filtering strategy as well as global feature alignment. The empirical results are better than several previous methods on multiple datasets.

**Questions:**

* How sensitive is the proposed method to the threshold of pseudo label filtering?


**Limitations:**

The authors did not clearly discuss limitations of their method.

**Strengths And Weaknesses:**

Strengths
* The categorization of test-time algorithms proposed in the paper is neat and convincing. Indeed, the evaluation protocol in test-time training and source-free adaptation has been sometimes inconsistent. The paper does a great job to streamline the evaluation protocols.
* The proposed anchored clustering also looks technically sound. While clustering, pseudo labelling and feature alignment have been separately explored in recent methods, the paper nicely connects them in a principled way. I also enjoy reading the detailed modifications of these components, e.g., eq 8.
* The empirical evaluation of the method looks complete and rigorous. It’s great to see that the proposed TTAC is compatible with other methods like SHOT and TTT++.

Weaknesses
* Many prior test-time algorithms work only under a narrow set of distribution shifts. I suspect the proposed method may be subject to some particular settings too. It would be good if the authors could clearly discuss limitations and failure cases.
* In particular, the proposed method introduces additional hyperparameters for peusde label filtering. It would be good to know how these parameters affect the performance.

---

> ### Author Response · Authors · 2022-08-02
> **Response to Reviewer Gh1A**
>
>
> We highly appreciate the invaluable comments and suggestions. In the response, we addressed the limitations and hyperparameter tuning. We hope the response could ease the concerns from the reviewer and earn firm support to this work.
>
> ### Limitations and Failure Cases
>
> We discuss the limitations of our method from two perspectives. First, we point out that TTAC implements backpropagation to update models at test stage, therefore additional computation overhead is required. Specifically, as requested by reviewer Af9y, we carried out additional evaluations on the per-sample wall clock time. Basically, we discovered that TTAC is twice computationally more expensive than BN and TENT but has the ability to strike a balance between efficiency and performance subject to the availability of computation resource. A more detailed analysis can be found in the response to reviewer Af9y.
>
> We further discuss the limitations on test-time training under more severe corruptions. Specifically, we evaluate TENT, SHOT and TTAC under 1-5 levels of corruptions on CIFAR10-C with results reported in Table 1. We observe generally a drop of performance from 1-5 level of corruption. Despite consistently outperforming TENT and SHOT at all levels of corruptions, TTAC's performance at higher corruption levels are relatively worse, suggesting more attention must be paid to more severely corrupted scenarios. We shall discuss the limitations in the revised manuscript.
>
> Table 1: Classification error under different levels of snow corruption on CIFAR10-C dataset.
>
> | Level |   1  |   2   |   3   |   4   |   5   |
> |:-----:|:----:|:-----:|:-----:|:-----:|:-----:|
> |  TEST | 9.46 | 18.34 | 16.89 | 19.31 | 21.93 |
> |  TENT | 8.76 | 11.39 | 13.37 | 15.18 | 13.93 |
> |  SHOT | 8.70 | 11.21 | 13.16 | 15.12 | 13.76 |
> |  TTAC | 6.54 |  8.19 |  9.82 | 10.61 |  9.98 |
>
>
> ### Sensitivity to Hyperparameters
>
> We evaluate the sensitivity to two thresholds during pseudo label filtering, namely the temporal smoothness threshold $\tau_{TC}$ and posterior threshold $\tau_{PP}$. $\tau_{TC}$ controls how much the maximal probability is allowed to deviate from the historical exponential moving average (ema). If the current value minus the ema is below a threshold, we believe the prediction is not confident and the sample should be excluded from estimating target domain cluster. $\tau_{PP}$ controls the the lower-bound of maximal probability and below this threshold is considered as not confident enough. We evaluate $\tau_{TC}$ in the interval between 0 and -1.0 and $\tau_{PP}$ in the interval from 0.5 to 0.95 with results on CIFAR10-C level 5 glass blur corruption presented in Table 2. We draw the following conclusions on the evaluations. i) There is a wide range of hyperparameters that give stable performance, e.g. $\tau_{PP}\in[0.5,0.0.9]$ and $\tau_{TC}\in[-0.01, -0.0001]$. ii) When temporal consistency filtering is turn off, i.e. $\tau_{TC}=-1.0$, because the probability is normalized to between 0 and 1, the performance drops substantially, suggesting the necessity to apply temporal consistency filtering.
>
>
> Table 2: Evaluation of pseudo labeling thresholds on CIFAR10-C level 5 glass blur corruption. Numbers are reported as classification error (%).
>
> | $\tau_{TC}\backslash \tau_{PP}$ |  0.5  |  0.6  |   0.7  |  0.8  |  0.9  |  0.95 |
> |:-------|:-----:|:-----:|:------:|:-----:|:-----:|:-----:|
> | **0.0**     | 23.03 | 22.26 |  21.96 | 22.50 | 21.14 | 28.55 |
> | **-0.0001** | 20.03 | 20.53 |  20.45 | 20.40 | 19.49 | 27.00 |
> | **-0.001**  | 19.66 | 20.51 |  19.49 | 20.48 | 19.42 | 26.83 |
> | **-0.01**   | 20.71 | 20.78 |  20.73 | 20.65 | 20.29 | 27.58 |
> | **-0.1**    | 24.10 | 21.47 |  21.46 | 22.36 | 21.45 | 28.71 |
> | **-1.0**    | 30.75 | 24.08 |  23.40 | 24.33 | 22.21 | 28.77 |

---

### Official Review · Reviewer_Af9y · 2022-07-17

**Rating:** 6
**Confidence:** 5
**Soundness:** 3 good
**Presentation:** 2 fair
**Contribution:** 3 good

**Summary:**

Test-time adaptation methods seek to improve generalization error from training on the source data to testing on the target data by updating the model on test data to mitigate shift between source and target.
This work re-examines the assumptions for this setting, and in particular identifies adapting _online_ in one pass and adapting _without_ altering the training loss as key considerations for practicality.
Multi-pass adaptation requires too much computation, delaying prediction, and modifying training requires re-training on all of the source data.
While arguing for these strict limits on the latency and the loss, this work relaxes the restriction to not use source data by allowing "light-weight" information (that is, statistics, clusters, etc.).
This variant of test-time training is named sequential test-time training (sTTT).

The test-time anchored clustering (TTAC) method is proposed to address this setting.
The anchored clustering is a joint clustering of the source and target, where the class-wise source clusters are the anchors, and the predicted test clusters are optimized to match the anchors in mean and variance by minimizing the KL divergence.
Both clusterings are computed as mixture-of-Gaussian distributions with one Gaussian per class.
The updates to the target clusters are regularized by exponential moving average (EMA) and stabilized by filtering out noisy predictions according to a queue of predictions on past test data (pseudo-label filtering).
Lastly, filtered out samples can still contribute to an update of the cluster statistics by global feature alignment.

Experiments show that TTAC rivals or improves on test-time adaptation and source-free adaptation baselines.
A variety of settings (one-pass/multi-pass and same/different training) and datasets (CIFAR-10/100-C, CIFAR-10.1, VisDA-C, and ModelNet-40-C) are evaluated.

**Questions:**

Questions

- Why is ImageNet-C not evaluated? How does TTAC perform on it? It is a gold standard benchmark, and results on CIFAR-10/100-C do not always reflect results at larger scale.
- How many updates are made when updating the network on the test input and sampling queue? Is there a single update per batch, as in TENT, or multiple updates like in TTT or MEMO?
- How much computation is needed for TTAC? Could you measure it in number of forward passes/backward passes or wallclock time per test sample? In particular, what is the overhead of the queueing?

Other Feedback

- Figure 1 is complex! To make it more accessible and informative, consider taking certain points and moving them into their own figure. For instance, PLF could perhaps be explained on its own.
- Note that SHOT has a journal edition in PAMI'21 with improved results. It is preferable to compare to the latest and strongest edition of the method.
- Here are a few typos noticed during review: "it is inevitable to access" on line 58 could be "still requires access to".
  On line 125 there is a missing period after "alone". On line 241 "Ours sTTT" should be "Our sTTT". There are more.


**Limitations:**

- There is no explicit discussion of limitations.
- The computational overhead of queuing data for statistics and parameter updates needs acknowledgement and ideally measurement. TTAC is more expensive than BN or TENT.

**Strengths And Weaknesses:**

Strengths

- This work helps explore a newer setting by emphasizing certain dimensions of the test-time adaptation problem, in particular (1) the use of source data, (2) online or one-pass vs. offline or multi-pass updates, and (3) altering training.
  As a newer setting, these choices need to be interrogated. The choice of sTTA in this work is different than some but not all prior work (in particular, BUFR [A] works in the same setting).
- The experiments explore a variety of settings (one-pass/multi-pass and same/different training) and datasets (CIFAR-10/100-C, CIFAR-10.1, VisDA-C, and ModelNet-40).
  Across these settings and datasets TTAC does as well or better than common baselines for test-time adaptation and source-free adaptation.
- The proposed TTAC method keeps improving more with additional data during testing compared to BN or T3A, which plateau at 20% of the test set.
- The use of source data does not alter training. It still requires the full source data for fitting the anchored clusters of TTAC offline, but fitting these clusters is agnostic to architecture, training loss, etc.
- Without source data, in the source-free adaptation setting, a variant of TTAC still works (Table 3) although its accuracy is close to that of existing methods (SHOT, TENT).
- The ablation study covers each component of the method and every setting considered.

Weaknesses

- sTTT is close to existing notions of test-time training, test-time adaptation, and source-free adaptation.
  sTTT underlines one-pass adaptation and minor use of source data, but test-time adaptation already focuses on one-pass adaptation _without_ source data, and several source-free methods report one-pass and multi-pass results (SHOT and BUFR, for example).
  A missing related paper, BUFR, has already argued for minimal use of source data---in their case, the collection of feature histograms---to improve adaptation peformance with results in the setting this work calls sTTT (see one-pass results).
- TTAC lacks technical novelty, as it applies several known components from source-free and test-time adaptation without a particularly strong point of innovation.
  - exponential moving average estimation: EATA [B], BN [C]
  - pseudo-label filtering: SHOT [18] has a prototypical clustering and pseudo-labeling filtering step that serves the same purpose as PLF
  - global feature alignment: TTT+ [19], as acknowledged in this work
  - queuing data to stabilize estimates: TTT+ [19], as acknowledged in this work
  - incremental mean and variance updates are well-known (Equation 8)
- The use of a queue for past data increases the computation for inference and adaptation. Algorithm 1 suggests the use of two devices, one for prediction and one for adaptation, but this is double the resources of fully test-time methods like BN or TENT.
- The experiments do not always align with the motivation and claims. In the fully source-free setting on CIFAR-10-C, the accuracy of TTAC is comparable to other methods like SHOT and TENT. Although sTTT emphasizes one-pass adaptation, Table 4 focuses on multi-pass adaptation, and results on VisDA-C in one pass are not reported.
  Although there is an emphasis on practicality and efficiency, results are not reported on ImageNet-C, which should be possible for an efficient method. It is potentially worrisome that most results are reported on the smallest dataset, CIFAR-10-C, and even on CIFAR-100-C the gap with existing methods shrinks (Table 1).
- (Minor) The writing needs editing throughout for word choice and grammar. (See "Other Feedback" in the next section for typos spotted.)


[A] BUFR: Source-Free Adaptation to Measurement Shift via Bottom-Up Feature Restoration. Eastwood et al. ICLR'22 and arXiv'21.

[B] EATA: Efficient Test-Time Model Adaptation without Forgetting. Niu et al. ICML'22 and arXiv'22 (~2 months before the NeurIPS deadline).

[C] BN: Improving robustness against common corruptions by covariate shift adaptation. Schneider et al. Neurips'20.

**Update**: I have raised my score given the benchmarking on ImageNet-C and profiling of computation time.

---

> ### Author Response · Authors · 2022-08-02
> **Response to Reviewer Af9y (part 4/4)**
>
> **Experiments.**
> Under the fully source-free protocol, TTAC is still better than SHOT. It is also worth noting that SHOT makes a hidden assumption that test domain is class balanced (through Eq.3 in the SHOT paper) and this assumption brings additional improvement to ours (TTAC+SHOT). For VisDA-C, we carried out additional evaluation in the one-pass protocol. As shown in Table 4, our TTAC still outperforms TENT and SHOT.
>
>
> Table 4: Classification error (%) on VisDA-C dataset under one-pass protocol.
>
> | Method | Plane | Bcycl |  Bus  |  Car  | Horse | Knife | Mcycl | Person | Plant | Sktbrd | Train | Truck | Per-class |
> |:------:|:-----:|:-----:|:-----:|:-----:|:-----:|:-----:|:-----:|:------:|:-----:|:------:|:-----:|:-----:|:---------:|
> |  TEST  | 56.52 | 88.71 | 62.77 | 30.56 | 81.88 | 99.03 | 17.53 |  95.85 | 51.66 |  77.86 | 20.44 | 99.51 |   65.19   |
> |  TENT  | 19.75 | 81.99 | 17.78 | 40.03 | 21.64 | 19.04 | 11.66 |  38.18 | 23.15 |  77.33 | 35.88 | 98.31 |   40.40   |
> |  SHOT  | 10.81 | 18.62 | 27.08 | 59.65 | 11.13 | 56.43 | 27.29 |  26.22 | 13.76 |  47.35 | 22.26 | 61.18 |   31.82   |
> |  TTAC  |  7.19 | 29.99 | 22.52 | 56.58 |  8.14 | 18.41 |  8.25 |  22.28 | 10.18 |  23.98 | 13.55 | 67.02 |   24.01   |
>
> ### Response to Other Feedbacks
>
> **Figure 1 complexity.**
> Thanks for the remind. We shall refine this illustration in the revised manuscript.
>
> **SHOT TPAMI21 version.**
> Thanks for referring to this paper. The journal version incorporated an additional self-supervised branch and focus on multi-pass source-free domain adaptation. We shall add discussion on this work in the revised manuscript.
>
> **Other Typos.**
> Thanks for spotting the typos. We shall revise in the final version.
>
> ### Discussion of limitations
>
> Thanks for suggesting the discussions on computational overhead. We shall add the above investigations into computation cost to the revised manuscript without any reservation.

---

> > ### Comment · Reviewer_Af9y · 2022-08-08
> > **Thank you for the response!**
> >
> > Thank you for providing new results, most importantly for benchmarking ImageNet-C and profiling computation, and for the point-by-point clarifications.
> >
> > The results on ImageNet-C, the gold standard benchmark at this time for test-time adaptation, and the discussion of computation resolved the two most important issues with the submission. As such I have raised my score from 3/Reject to 6/Weak Accept. This is a significant increase, but it is justified by the improved understanding of the proposed method given the rebuttal content. Of course, this content should definitely be incorporated into the paper if accepted—at a minimum the ImageNet-C table and even just a one sentence summary of the computational overhead (with a pointer to the supplement). I cannot increase the score any higher due to the increased computational cost (2x), but I now side with acceptance.

---

> > > ### Author Response · Authors · 2022-08-09
> > > **Thank you for the suggestions!**
> > >
> > > We would like thank the reviewer for supporting our work. We shall include the additional evaluations on ImageNet-C and wall clock time analysis in the revised manuscript and supplementary material. Again, we believe this work provides a timely overview and proper categorization of test time training works. The future works should be benchmarked under respective protocols with sTTT being one of the most realistic protocols.

---

> ### Author Response · Authors · 2022-08-02
> **Response to Reviewer Af9y (part 3/4)**
>
> ### Computation Cost Measured in Wall-Clock Time
>
> We highly appreciate the recommendation to measure wall-clock time as computation cost. We first measure the `overall wall time` as the time elapsed from the beginning to the end of test-time training, including all overheads. The per-sample wall time is then calculated as the `overall wall time` divided by the number of test samples. Experiments are carried out on the following platform, Ubuntu 20.04, Python 3.6.2, Pytorch 1.10.2, GeFore RTX 3090 and Intel(R) Xeon(R) Gold 5218 CPU @ 2.30GHz. We report the per-sample wall time (in seconds) for BN, TENT, SHOT and TTAC in Table 3 under different queue and update epoch settings. The `Inference` row indicates the per-sample wall time in a single forward pass including the data IO overhead. We make the following observations from the results. i) Maintaining a test sample queue introduces additional computation overhead for all methods, roughly in proportional to the size of test sample queue, while BN and TENT do not benefit from the additional resources. ii) Under the same experiment setting, BN and TENT are more computational efficient but TTAC is only twice more expensive than BN and TENT if no test sample queue is preserved (0.0083 v.s. 0.0030/0.0041). But the performance of TTAC w/o queue is still better than TENT (12.87 v
> .s. 13.48).  In summary, TTAC is able to strike a balance between computation efficiency and performance depending on how much computation resource is available. This suggests allocating a separate device is only necessary when securing best performance is the priority. We shall add the above studies and discussions in the revised manuscript.
>
> Table 3: The per-sample wall time (measured in seconds) on CIFAR10-C under sTTT protocol.
>
> (a) w/ Queue
>
> |  #Epochs  |    1   |    2   |    3   |    4   |
> |:---------:|:------:|:------:|:------:|:------:|
> |     BN    | 0.0136 | 0.0220 | 0.0293 | 0.0362 |
> |    TENT   | 0.0269 | 0.0399 | 0.0537 | 0.0663 |
> |    SHOT   | 0.0479 | 0.0709 | 0.0942 | 0.1183 |
> |    TTAC   | 0.0516 | 0.0822 | 0.1233 | 0.1524 |
> | Inference | 0.0030 | 0.0030 | 0.0030 | 0.0030 |
>
> (b) w/o Queue
>
> |  #Epochs  |    1   |
> |:---------:|:------:|
> |     BN    | 0.0030 |
> |    TENT   | 0.0041 |
> |    SHOT   | 0.0067 |
> |    TTAC   | 0.0083 |
> | Inference | 0.0030 |
>
> ### Clarifications on Other Misunderstandings
>
> **Relation to Other Protocols.**
> We notice that the test-time adaptation (TTA) proposed in TENT focuses on the one-pass protocol on the test data. Exploiting light-weight information from the source domain is not discussed in TTA protocol. Moreover, to the best of our knowledge, the source-free domain adaptation works, BUFR as mentioned by the reviewer, discussed an on-line adaptation protocol. But a thorough investigation and taxonomy is still missing in the community and our work fills the gap. We shall discuss BUFR in the revised manuscript.
>
> **Technical Novelty.**
> Thanks for discussing the technical relationship to existing works. We argue that the main technical novelty lies in aligning target clusters to the source ones which is not fully investigated by existing test-time training/adaptation works. Although pseudo labeling was adopted in SHOT, we notice that filtering out less noisy pseudo labels is vital to the success of TTAC, which has not been investigated by SHOT or other TTT works. We also discovered that using KL-Divergence for alignment has a better probabilistic explanation and outperforms L2 distance based objective widely adopted in TTT works.  Finally, the incremental estimation can substantially save memory footprint and is not well-known in the TTT community yet.
>
> **Queue Increases Computation.**
> The commonly perceived light-weight method, BN and TENT, are indeed more efficient. But TTAC without queue strikes a balance between efficiency and performance. If more computation resource is available, TTAC has a much higher upper bound than BN and TENT.

---

> ### Author Response · Authors · 2022-08-02
> **Response to Reviewer Af9y (part 2/4)**
>
> ### Test Sample Queue and Update Epochs
>
> The recommendation to evaluate the significance of test sample queue and update epochs is also highly appreciated.
> *First, throughout all experiments in the manuscript, we allow all competing methods under the N-O (sTTT) protocol to maintain the same test sample queue for fair comparison.* In the response, we further evaluate BN, TENT, SHOT and TTAC under a without queue protocol, i.e. only update model w.r.t. the current test sample batch. Second, we evaluate different numbers of model update epochs on the test sample queue for BN, TENT, SHOT and TTAC, fewer epochs means more computational efficient. The results on CIFAR10-C and ImageNet-C level 5 snow corruption are presented in Table 2. We make the following observations. i) Maintaining a sample queue can substantially improve the performance of methods that estimate target distribution, e.g. TTAC ($12.87\rightarrow10.88$ on CIFAR10-C) and SHOT ($15.18\rightarrow13.96$ on CIFAR10-C). This is due to more test samples giving a better estimation of true distribution. In contrast, BN and TENT update in a per-batch manner on the test sample queue and does not obviously benefit from observing more samples during the update.
> Next, we investigate different numbers of update epochs for BN, TENT, SHOT and TTAC. As seen from Table 2 (a) (c), for both CIFAR10-C and ImageNet-C datasets, TTAC and SHOT can consistently benefit from more update epochs on the test sample queue, in contrast, BN and TENT do not enjoy the same benefits. We ascribe this to iterative pseudo labeling benefiting from more update epochs. *In summary, if additional memory and computation is affordable at test time, TTAC can benefit from maintaining a test sample queue and more update epochs. Whilst, BN, TENT and SHOT can not fully unleash the power of additional computation resources at test-time training/adaptation.*
>
> Table 2: Comparing with and without test sample queue and different numbers of model update epochs. w/ Queue maintains a test sample queue with 4096 samples; w/o Queue maintains a single mini-batch with 256 and 128 samples on CIFAR10-C and ImageNet-C respectively.
>
> (a) CIFAR10-C w/ Queue
>
> | #Epochs | 1     | 2     | 3     | 4     |
> |:-------:|:-----:|:-----:|:-----:|:-----:|
> | BN      | 15.84 | 15.99 | 16.04 | 16.00 |
> | TENT    | 13.35 | 13.83 | 13.85 | 13.87 |
> | SHOT    | 13.96 | 13.93 | 13.83 | 13.75 |
> | TTAC    | 10.88 | 10.80 | 10.58 | 9.96  |
>
> (b) CIFAR10-C w/o Queue
>
> | #Epochs |   1   |
> |:-------:|:-----:|
> |    BN   | 15.44 |
> |   TENT  | 13.48 |
> |   SHOT  | 15.18 |
> |   TTAC  | 12.87 |
>
> (c) ImageNet-C w/ Queue
>
> | #Epochs |   1   |   2   |
> |:-------:|:-----:|:-----:|
> |    BN   | 62.34 | 62.34 |
> |   TENT  | 47.82 | 49.23 |
> |   SHOT  | 46.91 | 46.09 |
> |   TTAC  | 45.44 | 44.56 |
>
> (d) ImageNet-C w/o Queue
>
> | #Epochs |   1   |
> |:-------:|:-----:|
> |    BN   | 62.59 |
> |   TENT  | 48.39 |
> |   SHOT  | 51.46 |
> |   TTAC  | 47.02 |

---

> ### Author Response · Authors · 2022-08-02
> **Response to Reviewer Af9y (part 1/4)**
>
> Dear reviewer, we highly appreciate the comments and suggestions on experimental evaluations, analysis into computation cost and other concerns on overheads. In the response, we provide evaluations on ImageNet-C dataset, more in-depth analysis into computation cost and testing sample queue. We also highlighted the contributions on providing a categorization of TTT protocols such that follow-up works can make fair comparisons. We hope the additional evaluations and clarifications could ease the concerns and earn a full support to this work.
>
> ### Summary of Contributions
> We would like first to highlight the contributions of this work. There many recent works under the term of test-time training/adaptation. However, these works often rely on their own assumptions and as a result inappropriate comparisons are made frequently. Therefore, the first contribution we made is to classify the existing works based on i) whether source domain training objective must be changed or not and ii) whether sequential adaptation and inference is possible on the target domain. With the proposed criterion benchmarking different methods becomes more fair. Moreover, we advocate that sTTT is the most realistic protocol and propose TTAC based on the sTTT protocol. We demonstrate TTAC's superior results on all TTT protocols. **Therefore, we would like to highlight the contribution on categorizing TTT protocols and providing benchmarks under each protocol.** We believe that future works on TTT could inspire from our categorizations and focus more on fair comparison and realistic TTT protocols. **In terms of the TTT protocol, we do not claim the variants of sTTT have never appeared before, but instead propose a method, TTAC, as a solution with extensive benchmarking under the sTTT protocol as well as other protocols.** Finally, we notice that TTAC does not rely on any special network components, e.g. batch normalization adopted by BN and TENT, therefore **generalization to new architectures, e.g. transformer, where the special components no longer exist, is made possible.** Experiments with ViT backbone was presented in the original supplementary material.
>
> ### Additional Evaluation on ImageNet-C
> We highly appreciate the recommendation to evaluate on ImageNet-C dataset. In the response, we carried out experiments on ImageNet-C dataset under the N-O (sTTT) protocol. Comparisons with the state-of-the-art methods are presented in Table 1. We report the classification error (%) on severity 5 for the ImageNet-C dataset. We observe that our method (TTAC) outperforms BN and TENT with a large margin (5-13% improvement in error rate). Compared with SHOT, we are still ahead in average error rate and winning on 9 out of 15 types of corruptions. The results will be updated in the final manuscript.
>
>
> Table 1: Classification error (%) on ImageNet-C under the N-O (sTTT) protocol.
>
> | Method |  Birt | Contr | Defoc | Elast |  Fog  | Frost | Gauss | Glass | Impul |  Jpeg |  Motn | Pixel |  Shot |  Snow |  Zoom |  Avg  |
> |:------:|:-----:|:-----:|:-----:|:-----:|:-----:|:-----:|:-----:|:-----:|:-----:|:-----:|:-----:|:-----:|:-----:|:-----:|:-----:|:-----:|
> |  TEST  | 38.82 | 89.55 | 82.23 | 87.13 | 64.84 | 76.83 | 97.34 | 90.50 | 97.76 | 68.31 | 83.60 | 80.37 | 96.74 | 82.22 | 74.31 | 80.70 |
> |   BN   | 32.33 | 50.93 | 81.28 | 52.98 | 42.21 | 64.13 | 83.25 | 83.64 | 82.52 | 59.18 | 66.23 | 49.45 | 82.59 | 62.34 | 52.51 | 63.04 |
> |  TENT  | 31.39 | 40.27 | 75.68 | 42.03 | 35.38 | 64.32 | 84.92 | 84.96 | 81.43 | 46.84 | 49.48 | 39.77 | 84.21 | 49.23 | 43.49 | 56.89 |
> |  SHOT  | 30.69 | **37.69** | **61.97** | 41.30 | **34.74** | 54.19 | 76.33 | 71.94 | 74.24 | 46.50 | **47.98** | **38.88** | 70.60 | 46.09 | **40.74** | 51.59 |
> | TTAC   | **30.36** | 38.84 | 69.06 | **39.67** | 36.01 | **50.20** | **66.18** | **70.17** | **64.36** | **45.59** | 51.77 | 39.72 | **62.43** | **44.56** | 42.80 | **50.11** |

---

### Author Response · Authors · 2022-08-02
**Response to all reviewers**

We would like to highly appreciate all reviewers' efforts in providing valuable comments and constructive suggestions for improvement on our submission. First, we are very glad to see the positive comments by all reviewers. In particular, the contribution on **Categorization of TTT Protocols**, **Comprehensive Evaluation** and **Competitive Performance under All Protocols** are acknowledged by reviewers.
In this response, we shall address the questions over **Evaluation on Larger Dataset**, **Computation Overhead**, **Hyperparameter Tuning/Evaluations** and **Relation to Some Existing Works/Techniques**. Specifically, we evaluated ImageNet-C as an additional dataset, demonstrating superior performance. We also investigated the computation overhead and discovered that the TTAC is able to tradeoff between performance and computation cost. Additional evaluation on hyperparameter tuning suggests the robustness of TTAC. Relations to existing works are clarified in more details with experimental support. We hope the response can ease some of the concerns over novelty, insufficient experiments and hyperparameter tuning. Finally, we appreciate all reviewers again for considering this submission after revision for acceptance.

---

### Author Response · Authors · 2022-08-08
**Willing to Address Additional Comments**

Dear reviewers,

We would like to highly appreciate all reviewers' efforts and time again in providing valuable comments and constructive suggestions for improvement of our submission. We hope that the clarifications and additional evaluations provided in the responses have addressed all reviewers' questions and concerns.

We are always ready to provide additional clarifications should you have any questions and concerns during the discussion period, due on 9th Aug.

Thank you very much!

Authors

---

### Meta-Review · Area_Chair_fKmZ · 2022-08-28

**Recommendation:** Accept
**Confidence:** Certain

**Metareview:**

The paper proposes a clustering-based method for sequential test-time training, where the test data under distribution shifts arrives in a stream manner. The authors also explore a newer setting by emphasizing certain dimensions of the test-time adaptation problem: they distinguish experiment setups, from the perspective of whether it uses on-pass adaptation or multi-pass adaptation and whether it needs to alter the training phase. The distinction is important as prior works often compared under different setups, leading to unfair comparisons. The authors did a good job to address the reviewers' concerns in the author-reviewer discussion phase, and at the end, all reviewers unanimously support the acceptance. AC also did not find a particular weakness for rejection. As the test-time adaptation has emerged as a realistic solution to make ML models more robust to work, AC thinks that the contribution of this paper is of interest to a broad range of NeurIPS audience and would guide future research on the topic. Overall, AC is happy to recommend the acceptance.

**Award:**

No

---

### Decision · Program_Chairs · 2022-09-14

Accept